# Detecting Brittle Decisions for Free: Leveraging Margin Consistency in Deep Robust Classifiers

**Jonas Ngnawé**
IID-Université Laval and Mila
`jonas.ngnawe.1@ulaval.ca`

**Sabyasachi Sahoo**
IID-Université Laval and Mila
`sabyasachi.sahoo.1@ulaval.ca`

**Yann Pequignot**
IID-Université Laval
`yann.pequignot@iid.ulaval.ca`

**Frédéric Precioso**
Université Côte d'Azur, CNRS, INRIA, I3S, Maasai
`frederic.precioso@univ-cotedazur.fr`

**Christian Gagné**
IID-Université Laval, Mila and Canada CIFAR AI Chair
`christian.gagne@gel.ulaval.ca`

## Abstract

Despite extensive research on adversarial training strategies to improve robustness, the decisions of even the most robust deep learning models can still be quite sensitive to imperceptible perturbations, creating serious risks when deploying them for high-stakes real-world applications. While detecting such cases may be critical, evaluating a model's vulnerability at a per-instance level using adversarial attacks is computationally too intensive and unsuitable for real-time deployment scenarios. The input space margin is the exact score to detect non-robust samples and is intractable for deep neural networks. This paper introduces the concept of margin consistency – a property that links the input space margins and the logit margins in robust models – for efficient detection of vulnerable samples. First, we establish that margin consistency is a necessary and sufficient condition to use a model's logit margin as a score for identifying non-robust samples. Next, through comprehensive empirical analysis of various robustly trained models on CIFAR10 and CIFAR100 datasets, we show that they indicate high margin consistency with a strong correlation between their input space margins and the logit margins. Then, we show that we can effectively and confidently use the logit margin to detect brittle decisions with such models. Finally, we address cases where the model is not sufficiently margin-consistent by learning a pseudo-margin from the feature representation. Our findings highlight the potential of leveraging deep representations to assess adversarial vulnerability in deployment scenarios efficiently.

## 1 Introduction

Deep neural networks are known to be vulnerable to adversarial perturbations, visually insignificant changes in the input resulting in the so-called adversarial examples that alter the model's prediction (Biggio et al., 2013; Goodfellow et al., 2015). They constitute actual threats in real-world scenarios (Evtimov et al., 2017; Gnanasambandam et al., 2021), jeopardizing their deployment in sensitive and safety-critical systems such as autonomous driving, aeronautics, and health care. Research in the field has been intense and produced various adversarial training strategies to defend against the vulnerability to adversarial perturbations with bounded $\ell_p$ norm (e.g., $p = 2$, $p = \infty$) through

augmentation, regularization, and detection (Xu et al., 2017; Madry et al., 2018; Zhang et al., 2019; Carmon et al., 2019; Wang et al., 2020; Wu et al., 2020; Rice et al., 2020), to cite a few. The empirical robustness (adversarial accuracy) of these adversarially trained models is still far behind their high performance in terms of accuracy. It is typically estimated by assessing the vulnerability of samples of a given test set using adversarial attacks (Carlini & Wagner, 2016; Madry et al., 2018) or an ensemble of attacks such as the standard *AutoAttack* (Croce & Hein, 2020b). The objective of that evaluation is to determine if, for a given normal sample, an adversarial instance exists within a given $\epsilon$-ball around it. Yet, this robustness evaluation over a specific test set provides a global property of the model but not a local property specific to a single instance (Seshia et al., 2018; Dreossi et al., 2019). Beyond that specific test set, obtaining this information for each new sample would typically involve rerunning adversarial attacks or performing a formal robustness verification, which in certain contexts may be computationally prohibitive in terms of resources and time. Indeed, the computational cost makes it prohibitive to estimate robust accuracy at scale on large test sets and/or large models, for example, when using *AutoAttack* in standard mode. Moreover, in high-stakes deployment scenarios, knowing the vulnerability of single instances in real-time (i.e., their susceptibility to adversarial attacks) would be valuable, for example, to reduce risk, prioritize resources, or monitor operations. Therefore, there is a need for efficient and scalable ways to determine the vulnerability of a model's decision on a given sample.

The input space margin (i.e., the distance of the sample to the model's decision boundary in the input space), or input margin in short, can be used as a score to determine whether the sample is non-robust and, as such, likely to be vulnerable to adversarial attacks. Computing the exact input margin is intractable for deep neural networks (Katz et al., 2017; Elsayed et al., 2018; Jordan & Dimakis, 2020). These input margins may not be meaningful for fragile models with zero adversarial accuracies as all samples are vulnerable (close to the decision boundary). However, for robustly trained models, where only certain instances are vulnerable, the input margin is very useful for identifying the critical samples. Previous research studies have explored input margins of deep neural networks during training, focusing on their temporal evolution (Mickisch et al., 2020; Xu et al., 2023), and their exploitation in improving adversarial robustness through instance-reweighting with approximations (Zhang et al., 2020; Liu et al., 2021) and margin maximization (Elsayed et al., 2018; Ding et al., 2020; Xu et al., 2023). However, to the best of our knowledge, no previous research studies the relationship between the input space margin and the logit margin of robustly trained deep classifiers in the context of vulnerability detection.

In this paper, we investigate how the deep representation of robust models can provide information about the vulnerability of any single sample to adversarial attacks. We specifically address whether the logit margin as an approximation of the distance to the decision boundary in the feature space of the deep neural network (penultimate layer) can reliably serve as a proxy of the input margin for vulnerability detection. When this holds, we will refer to the model as being *margin-consistent*. The margin consistency property implies that the model can directly identify instances where its robustness may be compromised simply from a simple forward pass using the logit margin. Fig. 1 illustrates this idea of margin consistency. The following contributions are presented in the paper:

- We introduce the notion of *margin consistency*[1], a property to characterize robust models that allow the use of their logit margin as a proxy estimation for the input space margin in the context of non-robust sample detection. We prove that margin consistency is a necessary and sufficient condition to reliably use the logit margin for detecting non-robust samples.

- Through an extensive empirical investigation of pre-trained models on CIFAR10 and CIFAR100 with various adversarial training strategies, mainly taken from *RobustBench* (Croce et al., 2021), we provide evidence that almost all the investigated models display high margin consistency, i.e., there is a strong correlation between the input margin and the logit margin.

- We confirm experimentally that models with high margin consistency perform well in detecting samples vulnerable to adversarial attacks based on their logit margin. In contrast, models with weaker margin consistency exhibit poorer performance.

- For models where margin consistency does not hold, exhibiting a weak correlation between the input margins and the logit margins, we simulate margin consistency by learning to map the model's feature representation to a pseudo-margin with a better correlation through a simple learning scheme.

---

[1]Code available at: `https://github.com/ngnawejonas/margin-consistency`

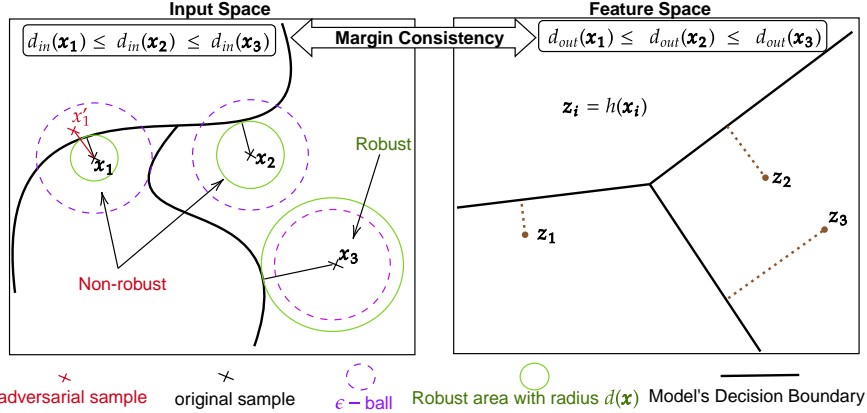

Figure 1: Illustration of the input space margin, margin in the feature space and margin consistency. The model preserves the relative position of samples to the decision boundary in the input space to the feature space.

## 2 Methodology

### 2.1 Notation and Preliminaries

**Notation** We consider $f_\theta : \mathbb{R}^n \to \mathbb{R}^K$ a deep neural network classifier with weights $\theta$ trained on a dataset of samples drawn iid from a distribution $\mathcal{D}$ on a product space $\mathcal{X} \times \mathcal{Y}$. Each sample $\mathbf{x}$ in the input space $\mathcal{X} \subset \mathbb{R}^n$ has a unique corresponding label $y \in \mathcal{Y} = \{1, 2, \ldots, K\}$. The prediction of $\mathbf{x}$ is given by $\hat{y}(\mathbf{x}) = \arg\max_{k \in \mathcal{Y}} f_\theta^k(\mathbf{x})$, where $f_\theta^k(\mathbf{x})$ is the $k$-th component of $f_\theta(\mathbf{x})$. We consider that a deep neural network is composed of a feature extractor $h_\psi : \mathcal{X} \to \mathbb{R}^m$ and a linear head with $K$ linear classifiers $\{\mathbf{w_k}, b_k\}$ such that $f_\theta^k(\mathbf{x}) = \mathbf{w_k}^\top h_\psi(\mathbf{x}) + b_k$. The predictive distribution $p_\theta(y|\mathbf{x})$ is obtained by taking the softmax of the output $f_\theta(\mathbf{x})$. A perturbed sample $\mathbf{x}'$ can be obtained by adding a perturbation $\delta$ to $\mathbf{x}$ within an $\epsilon$-ball $B_p(\mathbf{x}, \epsilon)$, an $\ell_p$-norm ball of radius $\epsilon > 0$ centered at $\mathbf{x}$, $\{\mathbf{x}' : \|\mathbf{x}' - \mathbf{x}\|_p = \|\delta\|_p < \epsilon\}$. The distance $\|\mathbf{x}' - \mathbf{x}\|_p = \|\delta\|_p$ represents the perturbation size defined as $(\sum_{i=1}^n |\delta_i|^p)^{\frac{1}{p}}$. In this paper, we will focus on $\ell_\infty$ norm ($\|\mathbf{x}\|_\infty = \max_{i=1,\ldots,n} |x_i|$), which is the most commonly used norm in the literature.

**Local robustness** Different notions of **local robustness** exist in the literature (Gourdeau et al., 2021; Zhong et al., 2021; Han et al., 2023). In this paper, we equate local robustness to $\boldsymbol{\ell_p}$**-robustness**, a standard notion corresponding to the invariance of the decision within the $\ell_p$ $\epsilon$-ball around the sample (Bastani et al., 2016; Fawzi et al., 2018) and formalized in terms of $\epsilon$-robustness.

**Definition 1.** *A model $f$ is $\epsilon$-robust at point $\mathbf{x}$ if for any $\mathbf{x}' \in B_p(\mathbf{x}, \epsilon)$ ($\mathbf{x}'$ in the $\epsilon$-ball around $\mathbf{x}$), we have $\hat{y}(\mathbf{x}') = \hat{y}(\mathbf{x})$.*

For a given robustness threshold $\epsilon$, a data instance is said to be non-robust for the model if this model is not $\epsilon$-robust on it. This means it is possible to construct an adversarial sample from that instance in its vicinity (i.e., within an $\epsilon$-ball distance from the original instance). A vulnerable sample to adversarial attacks is necessarily non-robust. This notion of local robustness can be quantified in the worst-case or, on average, inside the $\epsilon$-ball. We focus here on the worst-case measurement given by the input margin, also referred to as the *minimum distortion* or the *robust radius* (Szegedy et al., 2014; Carlini & Wagner, 2016; Weng, 2019)

**The input space margin** is the distance to the decision boundary of $f$ in the input space. It is the norm of a minimal perturbation required to change the model's decision at a test point $\mathbf{x}$:

$$d_{in}(\mathbf{x}) = \inf\{\|\delta\|_p : \delta \in \mathbb{R}^n \text{ s.t. } \hat{y}(\mathbf{x}) \neq \hat{y}(\mathbf{x} + \delta)\} = \sup\{\epsilon : f \text{ is } \epsilon\text{-robust at } \mathbf{x}\}. \quad (1)$$

An instance $\mathbf{x}$ is non-robust for a robustness threshold $\epsilon$ if $d_{in}(\mathbf{x}) \leq \epsilon$. Evaluating Eq. 1 for deep networks is known to be intractable in the general case. An upper bound approximation can be obtained using a point $\mathbf{x}'_0$, the closest adversarial counterpart of $\mathbf{x}$ in $\ell_p$ norm by $\hat{d}_{in}(\mathbf{x}) = \|\mathbf{x} - \mathbf{x}'_0\|_p$ (see Fig. 1).

**The logit margin** is the difference between the two largest logits. For a sample $\mathbf{x}$ classified as $i = \hat{y}(\mathbf{x}) = \arg\max_{j \in \mathcal{Y}} f_\theta^j(\mathbf{x})$ the logit margin is defined as $\left( f_\theta^i(\mathbf{x}) - \max_{j,j \neq i} f_\theta^j(\mathbf{x}) \right) > 0$. It is an approximation of the distance to the decision boundary of $f_\theta$ in the feature space. The decision boundary in the feature space around $\mathbf{z} = h_\psi(\mathbf{x})$, the feature representation of $\mathbf{x}$, is composed of $(K-1)$ linear decision boundaries (hyperplanes) $\mathrm{DB}_{ij} = \{\mathbf{z}' \in \mathbb{R}^m : \mathbf{w}_i^\top \mathbf{z}' + b_i = \mathbf{w}_j^\top \mathbf{z}' + b_j\}$ ($j \neq i$). The margin in the feature space is therefore the distance to the closest hyperplane $\min_{j,j \neq i} d(\mathbf{z}, \mathrm{DB}_{ij})$, where the distance $d(\mathbf{z}, \mathrm{DB}_{ij})$ from $\mathbf{z}$ to a hyperplane $\mathrm{DB}_{ij}$ has a closed-form expression:

$$d(\mathbf{z}, \mathrm{DB}_{ij}) = \inf\{\|\eta\|_p : \eta \in \mathbb{R}^m \text{ s.t. } \mathbf{z} + \eta \in \mathrm{DB}_{ij}\} = \frac{f_\theta^i(\mathbf{x}) - f_\theta^j(\mathbf{x})}{\|\mathbf{w}_i - \mathbf{w}_j\|_q}, \tag{2}$$

where $\|\cdot\|_q$ is the dual norm of $p$, $q = \frac{p}{p-1}$ for $p > 1$ (Moosavi-Dezfooli et al., 2016; Elsayed et al., 2018).

When the classifiers $\mathbf{w}_j$ are equidistant, *i.e.* $\|\mathbf{w}_i - \mathbf{w}_j\|_q = C$ whenever $i \neq j$, the margin becomes:

$$\min_{j,j \neq i} \frac{f_\theta^i(\mathbf{x}) - f_\theta^j(\mathbf{x})}{C} = \frac{1}{C} \min_{j,j \neq i} \left( f_\theta^i(\mathbf{x}) - f_\theta^j(\mathbf{x}) \right) = \frac{1}{C} \underbrace{\left( f_\theta^i(\mathbf{x}) - \max_{j,j \neq i} f_\theta^j(\mathbf{x}) \right)}_{\text{logit margin}}. \tag{3}$$

Under the equidistance assumption, the logit margin is proportional (equal up to a scaling factor) to the margin in the feature space. We will denote the logit margin of $\mathbf{x}$ by $d_{out}(\mathbf{x})$:

$$d_{out}(\mathbf{x}) = f_\theta^i(\mathbf{x}) - \max_{j,j \neq i} f_\theta^j(\mathbf{x}). \tag{4}$$

## 2.2 Margin Consistency

**Definition 2.** *A model is **margin-consistent** if there is a monotonic relationship between the input space margin and the logit margin, i.e., $d_{in}(\mathbf{x}_1) \leq d_{in}(\mathbf{x}_2) \Leftrightarrow d_{out}(\mathbf{x}_1) \leq d_{out}(\mathbf{x}_2), \forall \mathbf{x}_1, \mathbf{x}_2 \in \mathcal{X}$.*

A margin-consistent model preserves the relative position of samples to the decision boundary from the input space to the feature space. A sample further from (closer to) the decision boundary in the input space remains further from (closer to) the decision boundary in the feature space with respect to other samples, as illustrated in Fig. 1.

We can evaluate margin consistency by computing the **Kendall rank correlation** ($\tau \in [-1, 1]$) between the logit margins and the input margins over a test set. The Kendall rank correlation tests the existence and strength of a monotonic relationship between two variables. It makes no assumption on the distribution of the variables and is robust to outliers (Chattamvelli, 2024). While a positive value of $\tau$ indicates samples are ranked similarly (or identically for $\tau = 1$) according to logit margins and input margins, a negative value of $\tau$ indicates that one margin's ranking is roughly reversed. Perfect margin consistency corresponds to the situation $\tau = 1$.

## 2.3 Non-robust Samples Detection

Non-robust detection can be defined as a scored-based binary classification task where non-robust samples constitute the positive class, and the input margin $d_{in}$ induces a perfect discriminative function $g$ for that:

$$g(\mathbf{x}; f_\theta) = \mathbb{1}_{[d_{in}(\mathbf{x}) \leq \epsilon]} = \begin{cases} 1 & \text{if } \mathbf{x} \text{ is non-robust} \\ 0 & \text{if } \mathbf{x} \text{ is robust} \end{cases}.$$

If a model is margin-consistent, its logit margin can also be a discriminative score to detect non-robust samples. The following theorem establishes that this is a necessary and sufficient condition. Therefore, the degree to which a model is margin-consistent should determine the discriminative power of the logit margin.

**Theorem 1.** *If a model is margin-consistent, then for any robustness threshold $\epsilon$, there exists a threshold $\lambda$ for the logit margin $d_{out}$ that separates perfectly non-robust samples and robust samples. Conversely, if for any robustness threshold $\epsilon$, $d_{out}$ admits a threshold $\lambda$ that separates perfectly non-robust samples from robust samples, then the model is margin-consistent.*

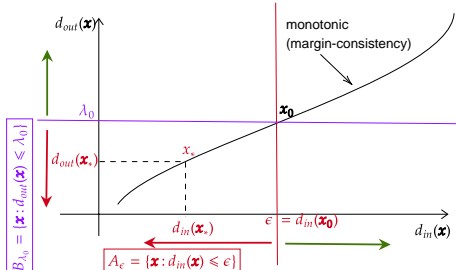
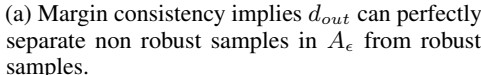
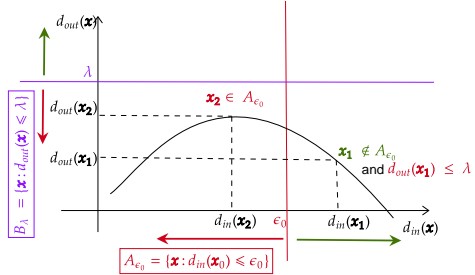

(a) Margin consistency implies $d_{out}$ can perfectly separate non robust samples in $A_\epsilon$ from robust samples.

(b) Without margin consistency, $d_{out}$ cannot be a good discriminator for robust and non-robust samples.

Figure 2: Illustration of Theorem 1's proof.

*Proof sketch.* Fig. 2 presents intuition behind the proof of Theorem 1. For the first part of the theorem (see Fig. 2a), if there is a monotonic relationship between $d_{in}$ and $d_{out}$ (margin consistency), any point $\mathbf{x}$ with $d_{in}$ less than the threshold $\epsilon$ (non-robust) will also have $d_{out}$ less than $\lambda_0 = d_{out}(\mathbf{x_0})$ (with $d_{in}(\mathbf{x_0}) = \epsilon$). For the second part (see Fig. 2b), if there are two points $\mathbf{x}_1$ and $\mathbf{x}_2$ with non-concordant $d_{in}$ and $d_{out}$ (no margin consistency), then for a threshold $\epsilon_0$ between $d_{in}(x_1)$ and $d_{out}(x_2)$, they will both have different classes but no threshold of $d_{out}$ can classify them both correctly. The complete proof of Theorem 1 is deferred to Appendix A. **Common metrics for detection** include (Hendrycks & Gimpel, 2017; Corbière et al., 2019; Zhu et al., 2023): the Area Under the Receiver Operating Curve (**AUROC**), which ensures the ability of a model to distinguish between the positive and negative classes across all possible thresholds; the Area Under the Precision-Recall Curve (**AUPR**), which evaluates the trade-off between precision and recall and is less sensitive to imbalance between positive and negative classes; and the False Positive Rate (FPR) at a 95% True Positive Rate (TPR) (**FPR@95**), that is crucial in systems where missing positive cases can have serious consequences, such as minimizing the number of vulnerable samples missed. The AUROC and AUPR of a perfect classifier is 1, while 0.5 for a random classifier.

# 3 Evaluation

## 3.1 Experimental Setup

**Datasets and models** We investigate various pre-trained models on CIFAR10 and CIFAR100 datasets (Krizhevsky, 2009). The majority of models were loaded from the *RobustBench* model zoo[2] (Croce et al., 2021), with a few more models that are ResNet-18 (He et al., 2016) models we trained on CIFAR10 with Standard Adversarial Training (Madry et al., 2018), TRADES (Zhang et al., 2019), Logit Pairing (ALP and CLP, Kannan et al. (2018)), and MART (Wang et al., 2020), using the experimental setup of Wang et al. (2020).

**Input margin estimation** This is done using FAB attack (Croce & Hein, 2020a), which is an attack that minimally perturbs the initial instance. Xu et al. (2023) used it in their adversarial training strategy as a reliable way to compute the closest boundary point given enough iterations. We perform the untargeted FAB attack without restricting the distortion to find the boundary for all the samples in the test set instead of constraining the perturbation inside a given $\epsilon$-ball when evaluating robustness. As a sanity check for the measured distances, we compare the ratio of correct samples $\mathbf{x}$ with estimated input margins greater than $\epsilon = 8/255$ and the robust accuracy in $\ell_\infty$ norm measured with *AutoAttack* (Croce & Hein, 2020b) at $\epsilon = 8/255$. Both quantities estimate the same thing, with a mean absolute difference over the models of $1.3$ and $0.48$ for CIFAR10 and CIFAR100, respectively, which are reasonable.

The estimation of the input margins over the $10,000$ test samples allows us to create for a given threshold $\epsilon$ a pool of vulnerable samples that can be successfully attacked at threshold $\epsilon$ and non-

---

[2]https://github.com/RobustBench/robustbench

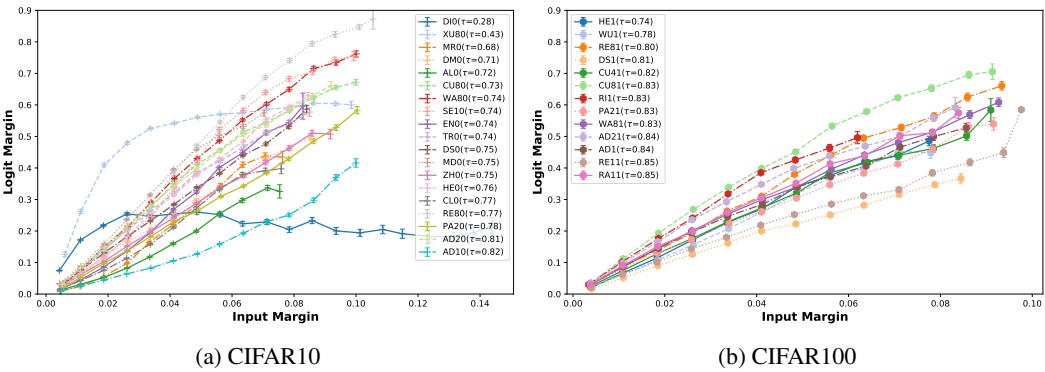

(a) CIFAR10          (b) CIFAR100

Figure 3: Margin consistency of various models: there is a strong correlation between input space margin and logit margin for most $\ell_\infty$ robust models tested, the exceptions being DI0 and XU80 on CIFAR10. See Table 1 for the references on the models. The correlations are given with standard error for the y-axis values in each interval.

vulnerable samples that were not able to be attacked. Training and distance estimations were run on an NVIDIA Titan Xp GPU (1x).

## 3.2 Results and Analysis

**Correlation analysis** The results presented in Fig. 3 show that the logit margin has a strong correlation (up to 0.86) with the input margin, which means that they have a level of margin consistency for those models. The plots are given with standard error for the y-axis values in each interval. However, we also observe that two models (i.e., **DI0** (Ding et al., 2020) and **XU80** (Xu et al., 2023) WideResNets) have a weaker correlation. We show in Sec. 3.3 that we can learn to map the feature representation of these models to a pseudo-margin that reflects the distance to the decision boundary in the input space. Additional results on ImageNet with $\ell_\infty$ norm and on $\ell_2$-robust models on CIFAR10 are given in Table 5 of appendix E.

**Vulnerable samples detection** We present the results for the robustness threshold $\epsilon = 8/255$ in Table 1. As expected with the strong correlations, the performance over the non-robust detection task is excellent. We can note that the metrics are lower for the two models with low correlations and particularly very high FPR@95. The performance remains quite good with different values of $\epsilon$ (cf. appendix B). Moreover, we show in appendix F.1 that the empirical robust accuracy of margin-consistent models can be accurately estimated by attacking only a small subset of the test set.

**Margin Consistency and Lipschitz Smoothness** A neural network $f$ is said to be $L$-Lipschitz if $\|f(\mathbf{x}_1) - f(\mathbf{x}_2)\| \leq L\|\mathbf{x}_1 - \mathbf{x}_2\|$, $\forall \mathbf{x}_1, \mathbf{x}_2$. Lipschitz smoothness is important for adversarial robustness because a small Lipschitz constant $L$ guarantees the network's output cannot change more than a factor $L$ of the change in the input. There are strategies to directly constraint the Lipschitz constant to achieve 1-Lipschitz networks (Cisse et al., 2017; Li et al., 2019; Serrurier et al., 2021; Araujo et al., 2023). Empirical adversarial training strategies only indirectly encourage Lipschitz's smoothness of the model. However, we note that Lipschitz's continuity of the feature extractor $h_\psi$ does not imply margin consistency of the model. Considering two points $\mathbf{x}_1$ and $\mathbf{x}_2$ with $0 < d_{in}(\mathbf{x}_1) < d_{in}(\mathbf{x}_2)$, the $L$-Lipschitz condition implies that $d_{out}(\mathbf{x}_i) \leq L d_{in}(\mathbf{x}_i)$ for $i = 1, 2$. However, as long as $d_{out}(\mathbf{x}_1) > 0$, it is possible *a priori* to have $d_{out}(\mathbf{x}_2) < d_{out}(\mathbf{x}_1)$, thus violating the margin consistency condition, while still satisfying the previous relations. We also note that the strength of the correlation, *i.e.* the level of margin consistency, does not depend on the robust accuracy (see Fig. 4a and 4b).

**Insight into when margin consistency may hold?** We hypothesize that when the feature extractor $h_\psi$ behaves locally as an isometry (preserving distances, up to a scaling factor $\kappa$, at least for directions normal to the decision boundary), i.e., $\|\mathbf{x} - \mathbf{x}'\|_p = \kappa\|h_\psi(\mathbf{x}) - h_\psi(\mathbf{x}')\|_p$, margin consistency will occur. Given an input sample $\mathbf{x}$, by definition $d_{out}(\mathbf{x}) = \|\mathbf{z} - \mathbf{z}'\|$ where $\mathbf{z} = h_\psi(\mathbf{x})$ and $\mathbf{z}'$ the closest point to $\mathbf{z}$ on the feature space decision boundary. The bijectivity of a local isometry implies that we have $h_\psi(\mathbf{x}') = \mathbf{z}'$, i.e. the representation of the closest point to $\mathbf{x}$ in input space matches the closest

| | Model ID | Kendall $\tau$ | AUROC | AUPR | FPR@95 | Acc | Rob. Acc | Architecture |
|---|---|---|---|---|---|---|---|---|
| CIFAR10 | **DI0** (Wu et al., 2020) | 0.28 | 67.49 | 70.91 | 82.56 | 84.36 | 41.44 | WideResNet-28-4 |
| | **XU80** (Xu et al., 2023) | 0.43 | 83.30 | 80.50 | 83.42 | 93.69 | 63.89 | WideResNet-28-10 |
| | **MR0** (Wang et al., 2020) | 0.68 | 92.95 | 94.92 | 29.76 | 79.69 | 39.12 | ResNet-18 |
| | **DM0** (Debenedetti et al., 2023) | 0.71 | 94.31 | 93.20 | 32.76 | 91.30 | 57.27 | XCiT-M12 |
| | **AL0** (Kannan et al., 2018) | 0.72 | 94.67 | 95.98 | 24.93 | 80.38 | 40.21 | ResNet-18 |
| | **CU80** (Cui et al., 2023) | 0.73 | 96.87 | 94.42 | 17.90 | 92.16 | 67.73 | WideResNet-28-10 |
| | **WA80** (Wang et al., 2023) | 0.74 | 96.82 | 94.33 | 17.60 | 92.44 | 67.31 | WideResNet-28-10 |
| | **SE10** (Sehwag et al., 2021) | 0.74 | 96.03 | 94.66 | 19.13 | 84.59 | 55.54 | ResNet-18 |
| | **EN0** (Engstrom et al., 2019) | 0.74 | 95.16 | 95.07 | 24.10 | 87.03 | 49.25 | ResNet-50 |
| | **TR0** (Zhang et al., 2019) | 0.74 | 94.63 | 96.13 | 30.93 | 80.72 | 42.23 | ResNet-18 |
| | **DS0** (Debenedetti et al., 2023) | 0.75 | 95.80 | 95.08 | 24.65 | 90.06 | 56.14 | XCiT-S12 |
| | **MD0** (Madry et al., 2018) | 0.75 | 95.36 | 97.00 | 23.23 | 81.85 | 36.91 | ResNet-18 |
| | **ZH0** (Zhang et al., 2019) | 0.75 | 95.86 | 95.65 | 24.91 | 84.92 | 53.08 | WideResNet-34-10 |
| | **HE0** (Hendrycks et al., 2019) | 0.76 | 96.35 | 95.68 | 20.01 | 87.11 | 54.92 | WideResNet-28-10 |
| | **CL0** (Kannan et al., 2018) | 0.77 | 95.93 | 96.98 | 20.01 | 81.12 | 40.08 | ResNet-18 |
| | **RE80** (Rebuffi et al., 2021) | 0.77 | 97.33 | 95.70 | 13.87 | 87.33 | 60.73 | WideResNet-28-10 |
| | **PA20** (Pang et al., 2022) | 0.78 | 97.65 | 96.39 | 14.40 | 88.61 | 61.04 | WideResNet-28-10 |
| | **AD20** (Addepalli et al., 2022) | 0.81 | 97.67 | 97.46 | 13.42 | 85.71 | 52.48 | ResNet-18 |
| | **AD10** (Addepalli et al., 2021) | 0.82 | 97.86 | 97.68 | 13.26 | 80.24 | 51.06 | ResNet-18 |
| CIFAR100 | **HE1** (Hendrycks et al., 2019) | 0.74 | 94.43 | 97.39 | 30.40 | 59.23 | 28.42 | WideResNet-28-10 |
| | **WU1** (Wu et al., 2020) | 0.78 | 95.81 | 98.00 | 23.34 | 60.38 | 28.86 | WideResNet-34-10 |
| | **RE81** (Rebuffi et al., 2021) | 0.80 | 96.87 | 98.30 | 18.06 | 62.41 | 32.06 | WideResNet-28-10 |
| | **DS1** (Debenedetti et al., 2023) | 0.81 | 96.78 | 98.30 | 19.18 | 67.34 | 32.19 | XCiT-S12 |
| | **CU41** (Cui et al., 2023) | 0.82 | 97.07 | 98.48 | 17.21 | 64.08 | 31.65 | WideResNet-34-10 |
| | **CU81** (Cui et al., 2023) | 0.83 | 97.41 | 98.24 | 15.62 | 73.85 | 39.18 | WideResNet-28-10 |
| | **RI1** (Rice et al., 2020) | 0.83 | 96.61 | 99.05 | 18.14 | 53.83 | 18.95 | PreActResNet-18 |
| | **PA21** (Pang et al., 2022) | 0.83 | 97.66 | 98.82 | 13.83 | 63.66 | 31.08 | WideResNet-28-10 |
| | **WA81** (Wang et al., 2023) | 0.83 | 97.51 | 98.28 | 14.96 | 72.58 | 38.83 | WideResNet-28-10 |
| | **AD21** (Addepalli et al., 2022) | 0.84 | 97.46 | 98.92 | 16.00 | 65.45 | 27.67 | ResNet-18 |
| | **AD1** (Addepalli et al., 2021) | 0.84 | 97.65 | 98.99 | 13.88 | 62.02 | 27.14 | PreActResNet-18 |
| | **RE11** (Rebuffi et al., 2021) | 0.85 | 97.97 | 99.05 | 13.21 | 56.87 | 28.50 | PreActResNet-18 |
| | **RA11** (Rade & Moosavi-Dezfooli, 2021) | 0.85 | 98.01 | 99.08 | 12.36 | 61.50 | 28.88 | PreActResNet-18 |

Table 1: Correlations and vulnerable points detection performance at $\epsilon = 8/255$ on different adversarially trained models.

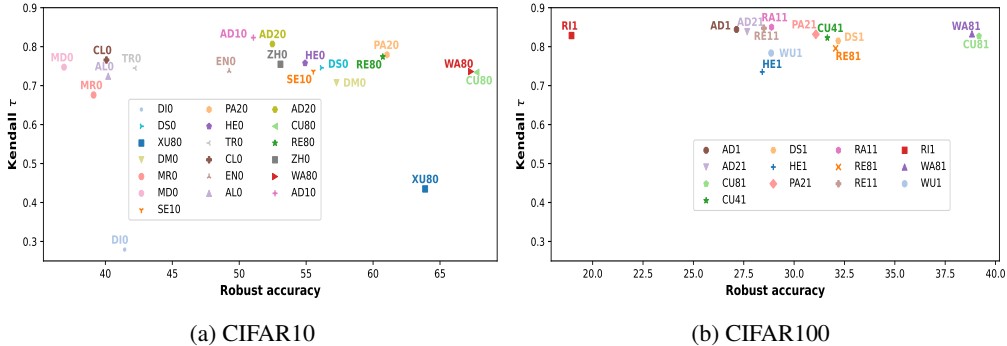

(a) CIFAR10          (b) CIFAR100

Figure 4: Distribution of the correlation between input margins and logit margins in $\ell_\infty$ with robust accuracy. The strength of the correlation, which indicates the level of margin consistency, does not depend on the robust accuracy. References on models are given in Table 1.

point to the representation of $\mathbf{x}$ in the feature space. In that case we will have $\|\mathbf{x} - \mathbf{x}'\| = \kappa\|\mathbf{z} - \mathbf{z}'\|$ which implies margin consistency. Experimentally, what we observe is that on the one hand, the input margin and the distance between the feature representations of $\mathbf{x}$ and $\mathbf{x}'$ (feature distance) correlate and on the other hand, the feature distance and the logit margin also correlate (Fig. 5a and Fig. 5b respectively).

## 3.3 Learning a Pseudo-Margin

For the two models that are weakly margin-consistent, we are proposing to directly learn a mapping that maps the feature representation of a sample to a pseudo-margin that reflects the relative position of the samples to the decision boundary in the input space. We use a learning scheme similar to the one of Corbière et al. (2019), with a small ad hoc neural network for learning the confidence of the instances. Given some samples with estimations of their input margins, the objective is to learn

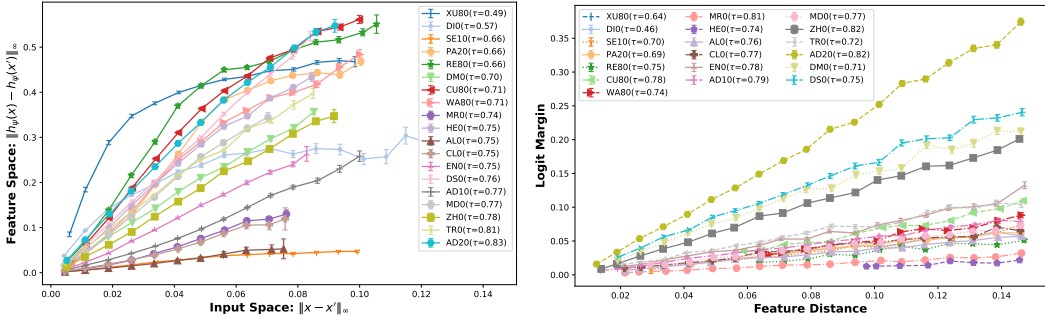

(a) Input space margins vs feature distances.

(b) Feature distances vs logit margins.

Figure 5: The correlations between the input margin, the distance between the feature representations of samples and their closest adversaries (feature distance $- \|h_\psi(x) - h_\psi(x')\|$), and the logit margin may be due to the local isometry of the feature extractor. See Table 1 for the specific references on the model ID. The correlations are given with standard error for the y-axis values in each interval.

to map their feature representation to a pseudo-margin that correlates with the input margins. This learning task can be seen as a learning-to-rank problem. We use a simple learning-to-rank algorithm for that purpose, which is a pointwise regression approach (He et al., 2008) relying on the mean squared error as a surrogate loss.

For the experiment, we used a similar architecture and training protocol as (Corbière et al., 2019) with a fully connected network with five dense layers of 512 neurons, with ReLU activations for the hidden layers and a sigmoid activation at the output layer. We learn using 5000 examples sampled randomly from the training set, with 20% (1000 examples) held as a validation. Fig. 6 and Table 2 show the improved correlation on the learned score compared to the logit margin for both models. The correlations are given with standard error for the y-axis values in each interval. The network has learned to recover the relative positions of the samples from the feature representation.

## 4   Related Work

**Detection tasks** in machine learning are found to be of three main types:

- **Adversarial Detection**   The goal of adversarial detection (Xu et al., 2017; Carlini & Wagner, 2017) is to discriminate adversarial samples from clean and noisy samples. An adversarial example is a malicious example found by adversarially attacking a sample; it has a different class while being close to the original sample. A vulnerable (non-robust) sample is a normal sample that admits an adversarial example close to it. The two detection tasks are very distinct. Adversarial detection is a defence mechanism like adversarial training; Tramer (2022) has established that both tasks are equivalent problems with the same difficulty.

- **Out-of-Distribution (OOD) detection**   In OOD detection (Hendrycks & Gimpel, 2017; Peng et al., 2024; Yang et al., 2021), the objective is to detect instances far from the distribution of the training data. These are often instances with different labels from the training labels or instances with the same label as training labels but with a covariate shift. For example, for a model trained on the CIFAR10 dataset, samples from the SVHN dataset (Netzer et al., 2011) or the corrupted version of CIFAR10-C (Hendrycks & Dietterich, 2019) are OOD samples for such a model.

- **Misclassification Detection (MisD)**   It consists in detecting whether the classifier's prediction is incorrect. This is also referred to as Failure Detection or Trustworthiness Detection (Corbière et al., 2019; Jiang et al., 2018; Luo et al., 2021; Granese et al., 2021; Zhu et al., 2023). MisD is often used for selective classification (classification with a reject option) (Geifman & El-Yaniv, 2017) to abstain from predicting samples on which the model is likely to be wrong. A score for non-robust detection cannot tell if the sample is incorrect, as a vulnerable sample could be from any side of the decision boundary.

**Formal robustness verification** aims at certifying whether a given sample is $\epsilon$-robust or if it is not an adversarial counter-example can be provided (Brix et al., 2023b). Some complete exact methods

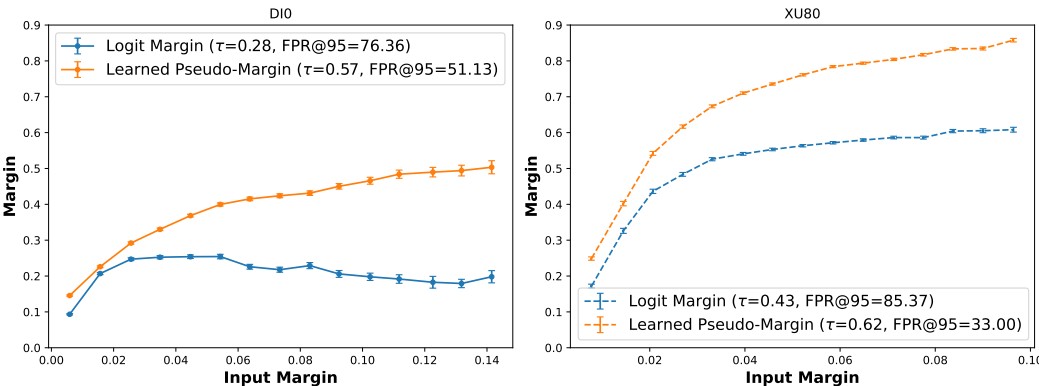

Figure 6: Correlation improvement of the learned pseudo-margin over the logit margin for DI0 (Ding et al., 2020) and XU80 (Xu et al., 2023).

| Model ID | Margin | Kendall tau (↑) | AUROC (↑) | AUPR (↑) | FPR@95 (↓) | Acc. | Rob. Acc |
|---|---|---|---|---|---|---|---|
| DI0 (Ding et al., 2020) | Logit margin | 0.28 | 67.49 | 70.91 | 82.56 | 84.36 | 41.44 |
| | Learned pseudo-margin | **0.57** | **88.49** | **89.04** | **51.13** | | |
| XU80 (Xu et al., 2023) | Logit margin | 0.43 | 83.30 | 80.50 | 83.42 | 93.69 | 63.89 |
| | Learned pseudo-margin | **0.62** | **93.66** | **90.22** | **33.00** | | |

Table 2: Comparison of the correlation and detection performance between the actual logit margin and the pseudo-margin learned. The models are initially weakly margin-consistent, but the pseudo-margin learned from feature representations simulates the margin consistency with higher correlation and better discriminative power.

based on solving Satisfiability Modulo Theory problems (Katz et al., 2017; Carlini et al., 2017; Huang et al., 2017) or Mixed-Integer Linear Programming (Cheng et al., 2017; Lomuscio & Maganti, 2017; Fischetti & Jo, 2017) provide formal certification given enough time. However, in practice, they are tractable only up to 100,000 activations (Tjeng et al., 2019). Incomplete but effective methods based on linear and convex relaxation methods and Branch-and-Bound methods (Zhang et al., 2018; Salman et al., 2019; Xu et al., 2020, 2021; Zhang et al., 2022; Shi et al., 2023) are faster but conservative, without guaranteed certifications even if given enough time. Scaling them to bigger architectures such as WideResNets and large Transformers is still challenging even with GPU accelartion(Brix et al., 2023a; König et al., 2024). Weng et al. (2018) converts the problem of finding the robust radius (input margin) as a local Lipschitz constant estimation problem. Computing the Lipschitz constant of Deep Nets is NP-hard (Virmaux & Scaman, 2018) and Jordan & Dimakis (2020) proved that there is no efficient algorithm to compute the local Lipschitz constant. The estimation provided by Weng et al. (2018) requires random sampling and remains computationally expensive to obtain a good approximation. Vulnerability detection with margin-consistent models does not provide certificates but an empirical estimation of the robustness of a sample as evaluated by adversarial attacks. At scale, it can help filter the samples to undergo formal verification and a more thorough adversarial attack for resource prioritization.

## 5 Limitations and Perspectives

**Vulnerability detection scope** The scope of this work is $\ell_p$ robustness measured by the input space margin; the minimum distortion that changes the model's decision while this does not give a full view of the $\ell_p$ robustness. Samples may be at the same distance to the decision boundary and have unequal unsafe neighbourhoods given by an average estimation over the $\epsilon$-neighbourhood considered. The average estimation of local robustness for a given $\epsilon$-neighborhood remains an open problem, so whether it is possible to extract other notions of robustness from the feature representation efficiently could be a potential avenue for further exploration.

**Attack-based verification** The margin consistency property does not rely on attacks; however, its verification and the learning of a pseudo-margin with an attack-based estimation may not be

possible if the model cannot be attacked on a sufficient number of samples. The assumption is that we can always successfully provide the closest point to the decision with a sufficient budget. This is a reasonable assumption since the studied models are not perfectly robust, and the empirical evidence so far with adaptive attacks is that no defence is foolproof, which justifies the need to detect the non-robust samples. It might occur that we need to combine with an attack such as *CW-attack* (Carlini & Wagner, 2016) to find the closest adversarial sample.

**Influence of terminal phase of training**    The work of Papyan et al. (2020) shows that when deep neural network classifiers are trained beyond zero training error and beyond zero cross-entropy loss (aka terminal phase of training), they fall into a state known as *neural collapse*. Neural collapse is a state where the within-class variability of the feature representations collapses to their class means, the class means, and the classifiers become self-dual and converge to a specific geometric structure, an equiangular tight frame (ETF) simplex, and the network classifier converges to nearest train class center. This implies that we may lose the margin consistency property. While neural collapse predicts that all representations collapse on their class mean, in practice, perfect collapse is not quite achieved, and it is precisely the divergence of a representation from its class mean (or equivalently its $\mathbf{w_i}$) which encodes the information we seek about the distance to the decision boundary in the input space. Exploring the impact of the neural collapse on margin consistency as models tend toward a collapsed state could provide valuable insights into generalization and adversarial robustness.

**Adaptaptive attacks and adversarial examples**    In this paper, we study the margin consistency of models on their training distribution by reporting the Kendall rank correlation between the logit margin and the input margin on the test set. The study of this property on inputs from a different distribution or specifically crafted examples is left for future research. However, we observe that the adversarial examples used for the input margin estimation have significantly smaller logit margins than the detection thresholds (see Table 4 in appendix D). This indicates that these specific adversarial examples are indeed identified as non-robust instances, together with clean non-robust samples.

# 6    Conclusion

This work addresses the question of efficiently estimating local robustness in the $\ell_p$ sense at a per-instance level in robust deep neural classifiers in deployment scenarios. We introduce margin consistency as a necessary and sufficient condition to use the logit margin of a deep classifier as a reliable proxy estimation of the input margin for detecting non-robust samples. Our investigation of various robustly trained models shows that they have high margin consistency, which leads to a high performance of the logit margins in detecting vulnerable samples to adversarial attacks. We also find that margin consistency does not always hold, with some models having a weak correlation between the input margin and the logit margin. In such cases, we show that it is possible to learn to map the feature representation to a better-correlated pseudo-margin that simulates the margin consistency and performs better on vulnerability detection. Finally, we present some limitations of this work, mainly the scope of robustness, the attack-based verification, the impact of neural collapse in terminal phases of training, and vulnerability to adaptive attacks. Beyond its highly practical importance, we see this as a motivation to extend the analysis of robust models and the properties of their feature representations in the context of vulnerability detection.

## Acknowledgements

This work is supported by the DEEL Project CRDPJ 537462-18 funded by the Natural Sciences and Engineering Research Council of Canada (NSERC) and the Consortium for Research and Innovation in Aerospace in Québec (CRIAQ), together with its industrial partners Thales Canada inc, Bell Textron Canada Limited, CAE inc and Bombardier inc.[3]

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

# A   Proof of Theorem 1

**Theorem 1.** *If a model is margin-consistent, then for any robustness threshold $\epsilon$, there exists a threshold $\lambda$ for the logit margin $d_{out}$ that perfectly separates non-robust samples and robust samples. Conversely, if for any robustness threshold $\epsilon$, $d_{out}$ admits a threshold $\lambda$ that perfectly separates non-robust samples from robust samples, then the model is margin-consistent.*

*Proof.* Formally, for a finite sample $S$ and nonegative values $\epsilon \geq 0$, $\lambda \geq 0$, we define:

$$A_\epsilon^S := \{x \in S : d_{in}(\mathbf{x}) \leq \epsilon\} \quad \text{and} \quad B_\lambda^S := \{x \in S : d_{out}(\mathbf{x}) \leq \lambda\}.$$

We say that $d_{out}$ perfectly separates non-robust samples from robust samples if for any finite sample $S \subseteq \mathcal{X}$ and every $\epsilon \geq 0$ there exists $\lambda \geq 0$ such that $A_\epsilon^S = B_\lambda^S$.

**Necessity**: We proceed by contraposition and assume that the model is not margin-consistent, i.e., there exist two samples $\mathbf{x}_1$ and $\mathbf{x}_2$ such that $d_{out}(\mathbf{x}_1) \leq d_{out}(\mathbf{x}_2)$ and $d_{in}(\mathbf{x}_1) > d_{in}(\mathbf{x}_2)$. By taking $S = \{\mathbf{x}_1, \mathbf{x}_2\}$ and $\epsilon = d_{in}(\mathbf{x}_2)$ we have that $A_\epsilon^S = \{\mathbf{x}_2\}$. However for any $\lambda \geq 0$, if $\mathbf{x}_2 \in B_\lambda^S$, then $d_{out}(\mathbf{x}_1) \leq d_{out}(\mathbf{x}_2) \leq \lambda$ and so $\mathbf{x}_1 \in B_\lambda^S$. Therefore $d_{out}$ does not perfectly separates non-robust samples from robust samples.

**Sufficiency**: Let's assume the model is margin-consistent. Let $S$ be a finite sample and consider a threshold $\epsilon$. Let $\mathbf{x}_0$ be an element of the finite set $A_\epsilon^S$ such that $d_{in}(\mathbf{x}_0) = \max\{d_{in}(x) : x \in A_\epsilon^S\}$ and let $\lambda = d_{out}(\mathbf{x}_0)$. Since the model is margin-consistent, then for every $\mathbf{x} \in S$:

$$x \in A_\epsilon^S \Leftrightarrow d_{in}(\mathbf{x}) \leq \epsilon \Leftrightarrow \underbrace{d_{in}(\mathbf{x}) \leq d_{in}(\mathbf{x}_0) \Leftrightarrow d_{out}(\mathbf{x}) \leq d_{out}(\mathbf{x}_0)}_{\text{margin consistency}} \Leftrightarrow d_{out}(\mathbf{x}) \leq \lambda \Leftrightarrow x \in B_\lambda^S.$$

This means we have $A_\epsilon^S = B_{\lambda_0}^S$, which shows that $d_{out}$ perfectly separates non-robust samples from robust samples. $\qquad\square$

Our formulation of perfect separation using finite samples is not fundamentally necessary, however it avoids dealing with the intricacy of the continuum.

# B   Detection Performance with Different Values of $\epsilon$

We present in Fig. 7 the performance of the detection for various values of the robustness threshold. We can see that the high margin consistency allows the logit margin to be a good proxy for detection at various thresholds. Note that below $\epsilon = 2/255$ and beyond $\epsilon = 16/255$, the ratio of vulnerable points to non-vulnerable points becomes too imbalanced, with little to no positive instances beyond $\epsilon = 32/255$.

# C   Equidistance Assumption of the Linear Classifiers

In Eq. 3 in Sec. 2.1, we show that we can approximate the exact feature margin by the logit margin when the classifiers $\mathbf{w}_k$ are equidistant, i.e. $\|\mathbf{w}_i - \mathbf{w}_j\| = C$ whenever $i \neq j$. The results in Table 3 show that using the logit margin instead of the exact minimum feature margin has a negligible effect on the results. However, computing the exact feature margin requires computing the minimum over $K - 1$ pairs of scaled logit differences. The approximation provided by the logit margin thereby circumvents the computational overhead of the minimum search, which can take a second instead of just microseconds for inference. This difference can add up to hours at scale, offering scalability when dealing with a large number of classes. We additionally provide boxplots for the $K(K - 1)/2$ distances between pairs of classifiers for each model (45 for CIFAR10 and 4950 for CIFAR100) in Fig. 8 and 9.

# D   Logit Margins of Adversarial Examples

Adversarial examples are perturbed samples that are close to the decision boundary. Therefore, we would expect these samples to have a very small logit margin for strongly margin-consistent models.

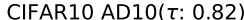

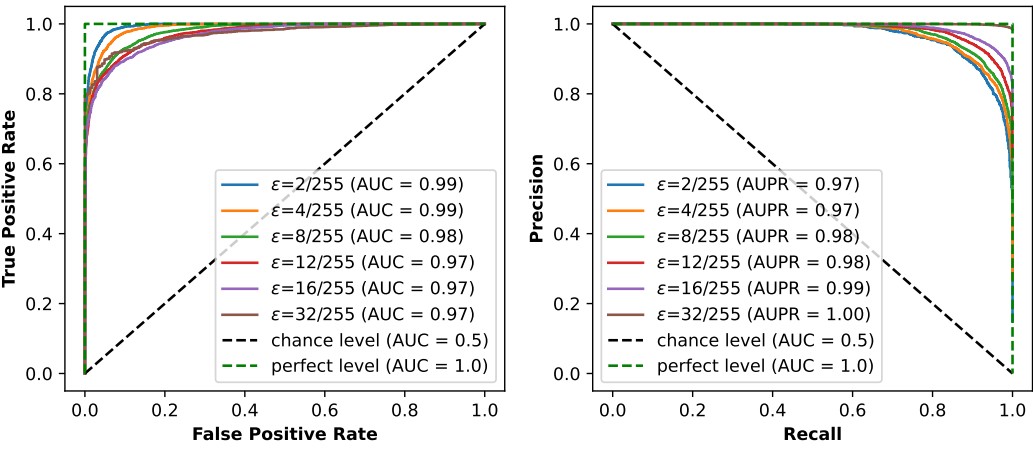

(a) Model **AD10** (Addepalli et al., 2021) on CIFAR10

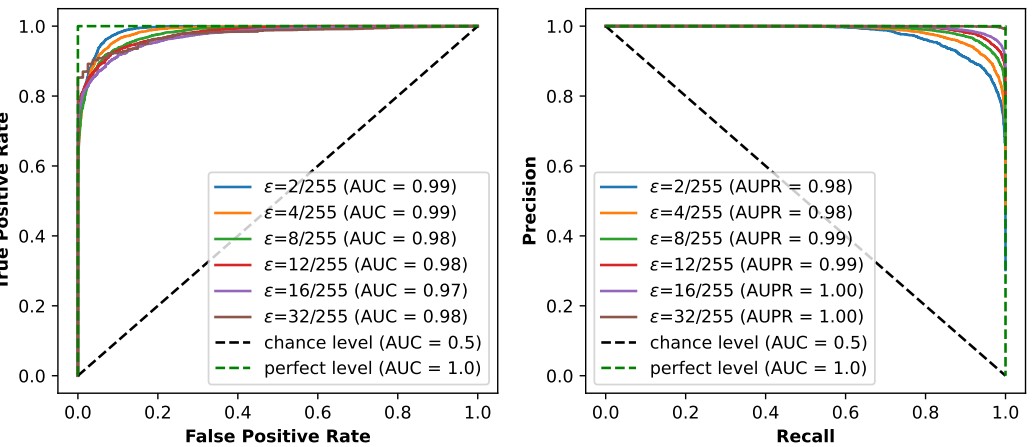

(b) **RA11** (Rade & Moosavi-Dezfooli, 2021) on CIFAR100

Figure 7: Variation of AUROC score for different threshold values $\epsilon$.

Here, we study the adversarial examples we used to estimate the input margin. In Table 4, we present the $99^{\text{th}}$ percentile of logit margins of adversarial examples and the detection threshold selected to obtain $95\%$ True Positive Rate. We can observe that the values of the adversarial logit margins are significantly smaller than the detection threshold, so they would be detected as non-robust – just like clean samples that lie close to the decision boundary.

# E   Results on ImageNet and $\ell_2$-robust models on CIFAR10

Table 5 shows the results with $\ell_\infty$-robust models on ImageNet (Deng et al., 2009), a larger dataset, and results on $\ell_2$-robust models (only available on CIFAR10 in *Robustbench*). The results extend well in both situations.

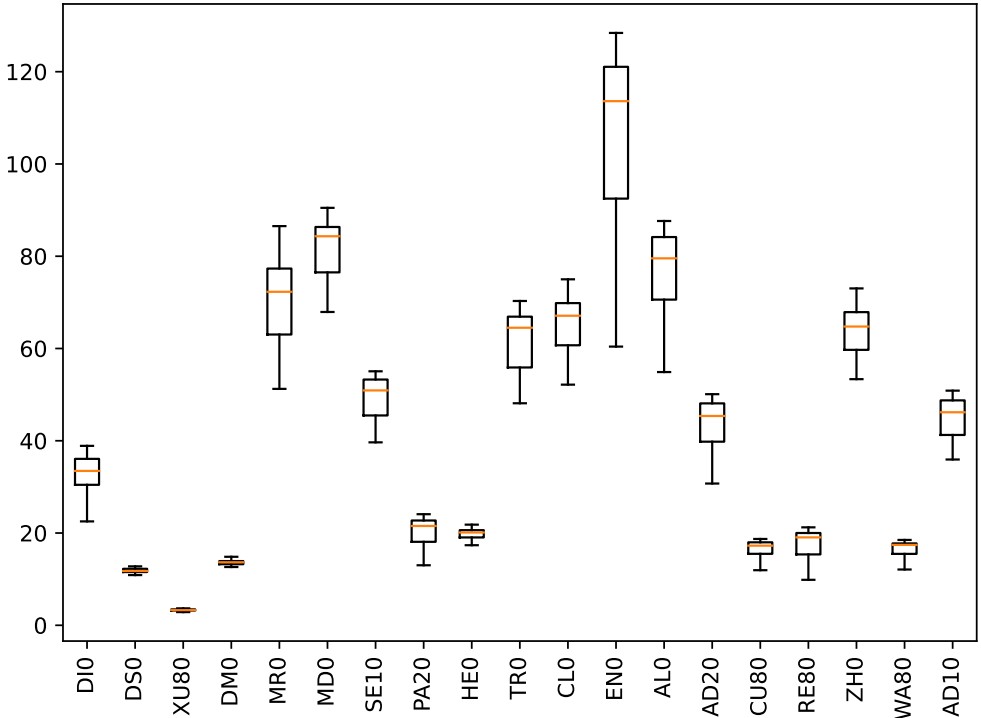

Figure 8: Equidistance of classifiers for CIFAR10 models. The boxplot reports the distances' minimum value, lower quartile (Q1), median, upper quartile (Q3), and maximum value.

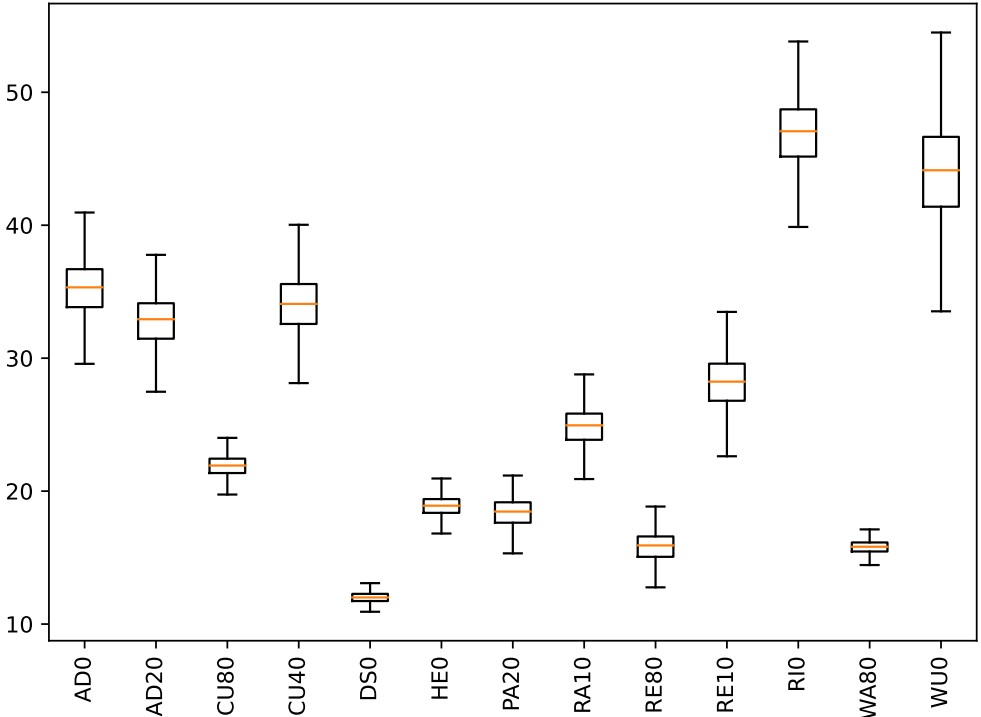

Figure 9: Equidistance of classifiers for CIFAR100 models. The boxplot reports the distances' minimum value, lower quartile (Q1), median, upper quartile (Q3), and maximum value.

| | Model | τ Lm | τ Fm | AUROC Lm | AUROC Fm | AUPR Lm | AUPR Fm | FPR95 Lm | FPR95 Fm | Acc | Rob. Acc | Architecture |
|---|---|---|---|---|---|---|---|---|---|---|---|---|
| CIFAR10 | DI0 | 0.28 | 0.32 | 67.49 | 70.75 | 70.91 | 74.28 | 82.56 | 80.57 | 84.36 | 41.44 | WideResNet-28-4 |
| | XU80 | 0.43 | 0.45 | 83.30 | 84.30 | 80.50 | 82.25 | 83.42 | 82.44 | 93.69 | 63.89 | WideResNet-28-10 |
| | MR0 | 0.68 | 0.70 | 92.95 | 93.73 | 94.92 | 95.54 | 29.76 | 27.43 | 79.69 | 39.12 | ResNet-18 |
| | AL0 | 0.72 | 0.74 | 94.67 | 95.12 | 95.98 | 96.28 | 24.93 | 22.21 | 80.38 | 40.21 | ResNet-18 |
| | CU80 | 0.73 | 0.75 | 96.87 | 97.52 | 94.42 | 95.47 | 17.90 | 14.77 | 92.16 | 67.73 | WideResNet-28-10 |
| | WA80 | 0.74 | 0.76 | 96.82 | 97.50 | 94.33 | 95.43 | 17.60 | 14.42 | 92.44 | 67.31 | WideResNet-28-10 |
| | SE10 | 0.74 | 0.74 | 96.03 | 96.59 | 94.66 | 95.52 | 19.13 | 16.85 | 84.59 | 55.54 | ResNet-18 |
| | EN0 | 0.74 | 0.76 | 95.16 | 95.90 | 95.07 | 95.85 | 24.10 | 21.22 | 87.03 | 49.25 | ResNet-50 |
| | TR0 | 0.74 | 0.77 | 94.63 | 95.52 | 96.13 | 96.74 | 30.93 | 25.73 | 80.72 | 42.23 | ResNet-18 |
| | DS0 | 0.75 | 0.75 | 95.80 | 95.93 | 95.08 | 95.23 | 24.65 | 23.34 | 90.06 | 56.14 | XCiT-S12 |
| | MD0 | 0.75 | 0.75 | 95.36 | 95.29 | 97.00 | 97.00 | 23.23 | 23.75 | 81.85 | 36.91 | ResNet-18 |
| | ZH0 | 0.75 | 0.77 | 95.86 | 96.35 | 95.65 | 96.14 | 24.91 | 21.74 | 84.92 | 53.08 | WideResNet-34-10 |
| | AD10 | 0.82 | 0.84 | 97.86 | 98.26 | 97.68 | 98.09 | 13.26 | 11.50 | 80.24 | 51.06 | ResNet-18 |
| CIFAR100 | HE1 | 0.74 | 0.74 | 94.43 | 94.41 | 97.39 | 97.39 | 30.40 | 30.40 | 59.23 | 28.42 | WideResNet-28-10 |
| | WU1 | 0.78 | 0.79 | 95.81 | 95.83 | 98.00 | 98.01 | 23.34 | 22.55 | 60.38 | 28.86 | WideResNet-34-10 |
| | RE812 | 0.80 | 0.80 | 96.87 | 96.91 | 98.30 | 98.32 | 18.06 | 17.49 | 62.41 | 32.06 | WideResNet-28-10 |
| | DS1 | 0.81 | 0.81 | 96.78 | 96.78 | 98.30 | 98.30 | 19.18 | 18.46 | 67.34 | 32.19 | XCiT-S12 |
| | CU41 | 0.82 | 0.82 | 97.07 | 97.09 | 98.48 | 98.49 | 17.21 | 17.35 | 64.08 | 31.65 | WideResNet-34-10 |
| | CU81 | 0.83 | 0.83 | 97.41 | 97.43 | 98.24 | 98.25 | 15.62 | 15.67 | 73.85 | 39.18 | WideResNet-28-10 |
| | RI1 | 0.83 | 0.83 | 96.61 | 96.62 | 99.05 | 99.06 | 18.14 | 17.70 | 53.83 | 18.95 | PreActResNet-18 |
| | PA21 | 0.83 | 0.83 | 97.66 | 97.70 | 98.82 | 98.84 | 13.83 | 13.80 | 63.66 | 31.08 | WideResNet-28-10 |
| | WA81 | 0.83 | 0.83 | 97.51 | 97.50 | 98.28 | 98.27 | 14.96 | 14.86 | 72.58 | 38.83 | WideResNet-28-10 |
| | AD21 | 0.84 | 0.84 | 97.46 | 97.48 | 98.92 | 98.93 | 16.00 | 15.36 | 65.45 | 27.67 | ResNet-18 |
| | AD1 | 0.84 | 0.84 | 97.65 | 97.67 | 98.99 | 98.99 | 13.88 | 13.34 | 62.02 | 27.14 | PreActResNet-18 |
| | RE11 | 0.85 | 0.84 | 97.97 | 97.88 | 99.05 | 99.01 | 13.21 | 13.36 | 56.87 | 28.50 | PreActResNet-18 |
| | RA11 | 0.85 | 0.85 | 98.01 | 98.01 | 99.08 | 99.08 | 12.36 | 12.20 | 61.50 | 28.88 | PreActResNet-18 |

Table 3: Correlations between the input margin and the logit margin (Lm) or the exact Feature margin (Fm) and detection scores on CIFAR10 ($\ell_\infty$, $\epsilon = 8/255$).

# F  Additional Applications

## F.1  Sample Efficient Robust Accuracy Estimation

Margin consistency enables empirical robustness evaluation over an arbitrarily large test set by only attacking a small subset of test samples. For a robustness evaluation at threshold $\epsilon$ (e.g., $\epsilon = 8/255$ in $\ell_\infty$ norm on CIFAR10 and CIFAR100), we randomly sample a small subset of the test set and determine the threshold $\lambda$ for the logit margin that best corresponds to $\epsilon$. We propose to use this threshold $\lambda$ to estimate the standard robust accuracy by evaluating the proportion of test samples which are correctly classified and whose logit margin is above $\lambda$ (see Algorithm 1). A naive way to set

---

**Algorithm 1** Sample Efficient Robustness Estimation with Margin Consistency

---

1: **Input:** Test Dataset $(X, Y) \in (\mathcal{X} \times \mathcal{Y})^N$, Robustness threshold $\epsilon > 0$, Subset size $n \ll N$.
2: **Output:** Robust Accuracy Estimation $\mathcal{A}_r$
3: - Select uniformly at random a subset $X_n$ of $n$ samples from $X$.
4: - Compute the estimations of the input margins on $X_n$, $D_n = \{\hat{d}_{in}(\mathbf{x}) : \mathbf{x} \in X_s\}$
5: - Create ground truth labels for vulnerability at threshold $\epsilon$ i.e. $\mathbb{1}_{[\hat{d}_{in}(\mathbf{x}) \leq \epsilon]}(\mathbf{x})$, for $\mathbf{x} \in X_s$.
6: - Determine the threshold $\lambda$ of $d_{out}$ that gives best approximation of robust accuracy on $X_s$.
7: - $\mathcal{A}_r = |\{\mathbf{x} \in X : d_{out}(\mathbf{x}) > \lambda \text{ and } \hat{y}(\mathbf{x}) = y\}|/N$

---

the threshold $\lambda$ at line 6 of Algorithm 1 would be to set it to the detection threshold at $\alpha = 95\%$ TPR or $\alpha = 90\%$ TPR, but the logit margin threshold could vary from one model to another; therefore a better way is to select it by tuning over values $\alpha \geq 0.80$ that gives the best approximation of the robust accuracy in terms of the absolute error on the small subset $X_s$. The same idea allows estimating a model's vulnerability over a large dataset without the labels.

We show that this leads to an accurate estimation of the robust accuracy of the investigated models evaluated with over $10,000$ by attacking only a random subset of size $500$. Fig. 10 shows the absolute error of the estimation obtained using $500$ samples. As expected, the estimation over the two weakly margin consistent models is not accurate while having a relatively small absolute difference on the strongly margin consistent models.

## F.2  Robustness Bias Analysis using the Logit Margin

Robust models often display robustness bias, namely a disparity of robustness across classes. Interestingly, in a strongly margin-consistent model (see Fig. 11, top row), we show that these discrepancies across classes with respect to input margin are reflected in the logit margin. Additionally, the margin

| | Model ID | Logit margin of Adversarial Examples (99th percentile) | Threshold of logit margin at 95% TPR for detection |
|---|---|---|---|
| CIFAR10 | AD1 | 0.01 | 1.42 |
| | DS0 | 0.61 | 2.19 |
| | DM0 | 1.64 | 3.01 |
| | MR0 | 0.04 | 8.35 |
| | MD0 | 0.06 | 11.04 |
| | SE10 | 0.06 | 2.83 |
| | PA20 | 0.14 | 1.69 |
| | HE0 | 0.07 | 2.82 |
| | TR0 | 0.07 | 4.11 |
| | CL0 | 0.03 | 5.76 |
| | EN0 | 0.13 | 5.06 |
| | AL0 | 0.03 | 5.11 |
| | AD20 | 0.03 | 1.33 |
| | CU80 | 1.32 | 2.02 |
| | RE802 | 0.13 | 1.87 |
| | ZH0 | 0.17 | 2.87 |
| | WA80 | 0.70 | 2.14 |
| | AD10 | 0.01 | 1.20 |
| CIFAR100 | AD1 | 0.01 | 1.42 |
| | AD21 | 0.02 | 1.44 |
| | CU81 | 0.13 | 2.09 |
| | CU41 | 0.04 | 1.95 |
| | DS1 | 0.13 | 1.36 |
| | HE1 | 0.06 | 1.81 |
| | PA21 | 0.08 | 1.56 |
| | RA11 | 0.08 | 1.77 |
| | RE812 | 0.17 | 1.73 |
| | RE11 | 0.07 | 1.40 |
| | RI1 | 0.03 | 3.42 |
| | WA81 | 0.09 | 1.59 |
| | WU1 | 0.06 | 2.02 |

Table 4: Logits of the adversarial examples are very small compared to the logit margin for the logit margin threshold for the detection at epsilon=8/255. See Table 1 for the specific references on the model ID.

| | Model ID | Kendall tau | AUROC | AUPR | FPR95 | Acc | Rob. Acc | Architecture |
|---|---|---|---|---|---|---|---|---|
| CIFAR10 ($\ell_2$) | DI02 | 0.66 | 95.91 | 93.05 | 21.94 | 88.02 | 66.09 | WideResNet-28-4 |
| | AU02 | 0.69 | 95.59 | 87.41 | 19.30 | 91.08 | 72.91 | ResNet-50 |
| | SE102 | 0.72 | 97.07 | 90.99 | 13.25 | 89.76 | 74.41 | ResNet-18 |
| | WA802 | 0.74 | 98.55 | 93.59 | 8.00 | 95.16 | 83.68 | WideResNet-28-10 |
| | RI02 | 0.74 | 96.82 | 92.86 | 13.98 | 88.67 | 67.68 | PreActResNet-18 |
| | EN02 | 0.75 | 96.92 | 92.78 | 13.25 | 90.83 | 69.24 | ResNet-50 |
| | RO02 | 0.76 | 98.19 | 96.51 | 10.15 | 89.05 | 66.44 | WideResNet-28-10 |
| | RE802 | 0.77 | 98.72 | 95.02 | 6.66 | 91.79 | 78.80 | WideResNet-28-10 |
| | RE102 | 0.79 | 98.58 | 95.32 | 7.04 | 90.33 | 75.86 | PreActResNet-18 |
| | RA102 | 0.80 | 99.00 | 96.78 | 5.53 | 90.57 | 76.15 | PreActResNet-18 |
| ImageNet ($\ell_\infty$) | SSI0 | 0.63 | 90.95 | 90.06 | 36.47 | 72.56 | 48.08 | ViT-S + ConvStem |
| | ENI0 | 0.73 | 94.95 | 97.34 | 25.07 | 62.56 | 29.22 | ResNet-50 |
| | SAI1 | 0.74 | 95.00 | 96.50 | 24.88 | 64.02 | 34.96 | ResNet-50 |
| | SAI0 | 0.77 | 95.16 | 97.99 | 29.31 | 52.92 | 25.32 | ResNet-18 |
| | WOI0 | 0.78 | 96.28 | 98.34 | 25.65 | 55.62 | 26.24 | ResNet-50 |

Table 5: Correlations between the input margin and the logit margin and detection scores on CIFAR10 ($\ell_2, \epsilon = 0.5$) and ImageNet ($\ell_\infty, \epsilon = 4/255$, 1000 samples). See Table 1 for the specific references on the model ID.

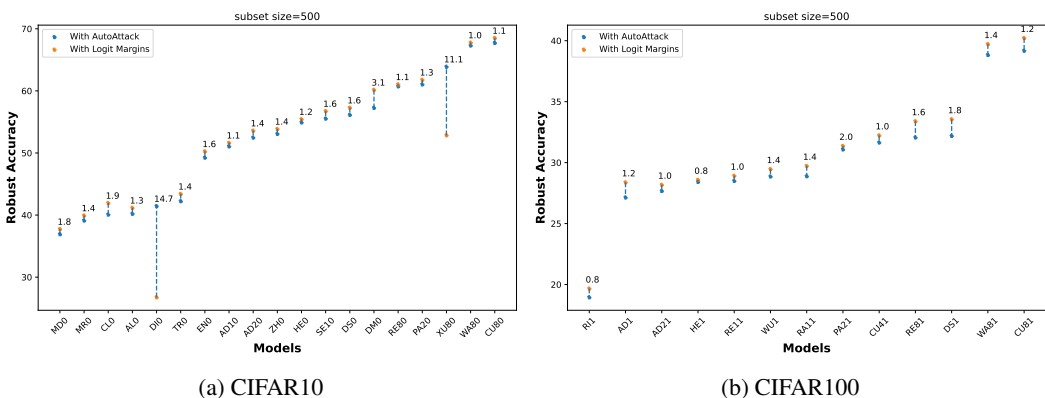

|                        |                        |
| :--------------------: | :--------------------: |
| (a) CIFAR10            | (b) CIFAR100           |

Figure 10: Estimations of the robust accuracy reported by *Robustbench* using logit margins with only 500 samples are quite accurate both on CIFAR10 and CIFAR100 for strongly margin-consistent models. The numbers indicate the absolute difference between the two values, averaged over ten subsets. See Table 1 for the specific references on the model ID.

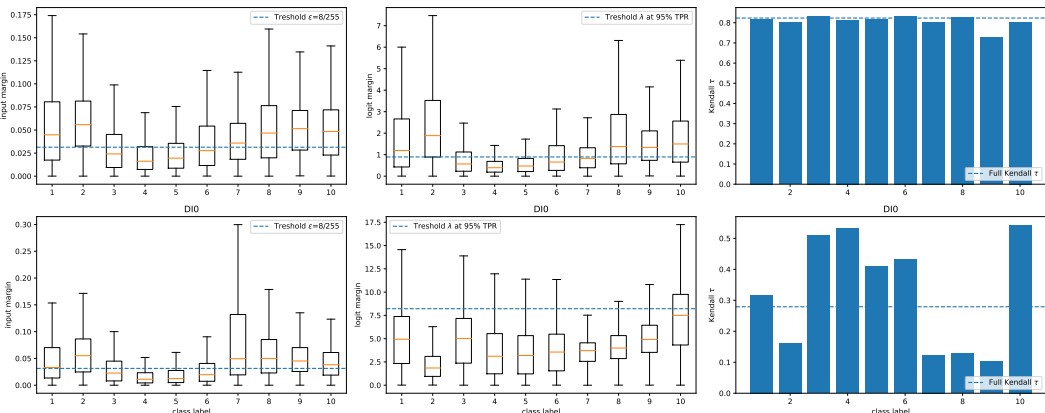

Figure 11: Robustness bias analysis with the input margin (first column) vs the logit margin (second column) and correlation scores per class (third column). We have a margin-consistent model on the top row, and on the bottom row, we have a weakly margin-consistent model. On the first column (resp. second column), each boxplot represents the distribution of the input margin (resp. logit margin) for the corresponding CIFAR10 class. The dotted blue line indicates the threshold on the first two columns and is the correlation score computed over all classes. The threshold $\lambda$ in the second column is the logit margin threshold at 95% TPR for detection at $\epsilon = 8/255$. The robustness bias of the strongly margin consistent model can be detected using the logit margin, unlike the weakly margin consistent model.

consistency remains strong for each class. However, for a weakly margin-consistent model (Fig. 11, bottom row), significant disparities exist between the correlations across classes, making using logit margin as a proxy for input margin problematic.

