# OpenReview forum: "Detecting Brittle Decisions for Free: Leveraging Margin Consistency in Deep Robust Classifiers"
_NeurIPS.cc/2024/Conference — NeurIPS 2024 poster_

### Official Review · Reviewer_WVo4 · 2024-07-06

**Soundness:** 2
**Presentation:** 3
**Contribution:** 2
**Rating:** 5
**Confidence:** 4

**Summary:**

This paper discovers that many existing neural network classifiers share a strong correlation between input margin (distance to decision boundary) and output margin (difference between top-2 logits). The paper further proposes to use this property for local robustness estimation.

**Strengths:**

Efficient local robustness estimation is an important problem, and the proposed method offers a viable direction for this purpose. The strong correlation between input and output margins is an interesting discovery. Furthermore, the paper clearly formalizes and quantifies this correlation via Kendall rank correlation. The paper is well-written, typo-free, and clearly conveys the results.

**Weaknesses:**

The weaknesses are mainly twofold.

First, the observation of the correlation, while interesting, is not strong enough in my opinion. Given that robust neural networks are known to be Lipschitz [1], the correlation between input and output margins is not very surprising. The main theoretical result (Theorem 1) is relatively straightforward, self-explanatory, and unsurprising.

Second, the authors did not consider the possibility of adaptive attacks. While the correlation between input and output margins is strong when the input itself is benign, it is unclear whether this correlation still holds when the input itself is adversarial. Is it possible to use attack methods to find inputs that break this correlation (i.e., output margin is large but input margin is small), leading to overestimation of the local robustness? Furthermore, the pseudo-margin prediction model adds even more potential vulnerabilities.

[1] Yang et al. A closer look at accuracy vs. robustness.

**Questions:**

- Equation (3) relies on an *equidistant* assumption. Is this assumption satisfied by real-world models? I found some short discussions in Appendix C, but would prefer a more formal analysis. Furthermore, Line 640 says "the values vary only in a small range." What do the "values" refer to? It also seems to me that Figures 8 and 9 demonstrate non-trivial variance?

- In addition to observing the correlation between input and output margins, the paper also mentions the distances in the feature space. How is the property regarding feature-space distance used in practice? For context, comparing Figure 5a and 5b, it seems that the input-output margin correlation is stronger than the input-feature margin correlation.

**Limitations:**

See weaknesses.

---

> ### Author Rebuttal · Authors · 2024-08-04
>
> Thank you for your review! Please find below our responses to your concerns.
>
> **a/ Lipschtz Smoothness vs. Margin Consistency**
>
> Thank you for raising this point. Lipschitz smoothness is an essential property of robust models since a small Lipschitz constant $L$ guarantees the network's output cannot vary more than a factor $L$ of the input variation. Empirical adversarial training strategies aim in a certain way to achieve Lipschitz's smoothness indirectly with various levels of Lipschitz constants, and it is also possible to directly constraint the Lipschitz constant to achieve $1$-Lipschitz networks ([1] and references therein). However, *Lipschitz smoothness does not imply margin consistency*. Indeed:
>
> *Let $f$ be an $L$-Lipschitz neural network i.e. $||f(x_1)-f(x_2)|| \leq L||x_1-x_2||$, $\forall x_1, x_2$. Let us consider two points $x_1$ and $x_2$ with $0<d_{in}(x_1)<d_{in}(x_2)$. Note that the $L$-Lipschitz condition implies that $d_{out}(x_i)\leq L d_{in}(x_i)$ for $i=1,2$. However, as long as $d_{out}(x_1)>0$, it is clearly possible a priori to have $d_{out}(x_2)<d_{out}(x_1)$, thus violating the margin consistency condition, while still satisfying the previous relations.*
>
> This theoretical possibility is supported by the empirical evidence that we found two robust (hence probably Lipschitz)  models which are not margin-consistent. The question of how both properties influence each other is out of the scope of this work and may be grounds for future work. We propose to add this clarification to the paper.
>
> The theorem itself establishes that margin consistency (preservation in the feature space, of the order of the samples' input space distances to the decision boundary) is a necessary and sufficient condition for using the logit margin as a proxy score for the input margin.
>
>
> **b/ Possibility of adaptive attacks**
>
> Implicitly, margin consistency, like standard accuracy, measures a property of the model over test samples iid with the training data. We leverage it to detect vulnerable iid test samples susceptible to being adversarially attacked. We concede that it may indeed be possible to craft examples that are non-robust ($d_{in} \leq \epsilon)$) but for which $d_{out}>\lambda$, and therefore would bypass the proposed detector. We believe studying the margin consistency on adversarial examples or designing adaptive attacks to be out of the scope of this work.
>
> **c/ (Q1) Equidistance Assumption**
>
> In Appendix C, by "values," we refer to the distances between the classifiers $||w_i-w_j||_q$ and by a small range, we mean that the interquartile range is small compared to the mean value of the distance. The empirical evidence shows that using the logit margin instead of the exact feature margin (minimum over margins to other classes) marginally affects the results and avoids the computational overhead of searching the minimum (See the general response section and Table 1 of the attached PDF file).
>
> **d/  (Q2) The property regarding the feature space distance**
>
> In Figure 5, by the feature space distance, we mean the distance $||h(x)-h(x’)||$ in the feature space between the representations of a sample $x$ and the representation of its adversarial example $x’$. Although the correlation is a bit stronger than with the logit margin, it is as hard to compute as the input margin ($||x-x’||$) since they both require finding the closest adversarial example in the input space.
>
> [1] Araujo, Alexandre, et al. "A unified algebraic perspective on Lipschitz neural networks." arXiv preprint arXiv:2303.03169 (2023).

---

> ### Comment · Reviewer_WVo4 · 2024-08-11
> **Thank you for the response**
>
> Thank you for the response. While the response cleared up my confusion on Q2, I still have some remaining questions.
>
> - (Minor) I agree that Lipschitzness and margin consistency are different concepts and it is possible to find counter-examples that satisfy one but not the other. That being said, they are still intuitively and empirically connected. The authors found two models that are "likely Lipschitz" but not margin-consistent, but also found 17 models that are robust margin-consistent (Figure 2). Hence, the theoretical results, while valid, are unsurprising in my opinion.
>
> - (Major) As mentioned in multiple places in the paper, this work focuses on detecting non-robust decisions with logit margins, which often manifest as adversarial examples. If it is possible to construct adversarial examples that can break the margin consistency and deceive the logit margin-based detection, then the purpose of the proposed method is defeated.
>
> ***Hence, I kindly disagree that studying the margin consistency on adversarial examples or designing adaptive attacks is out of the scope of this work, and believe that they must be carefully analyzed***. This is especially important because existing work has shown that adaptive attacks can significantly compromise adversary detectors [1].
>
> - (Minor) I am still confused about Figures 8 and 9. The authors mentioned that the distances between the classifiers vary by a small range. However, the data points in Figures 8 and 9 seem all over the place.
>
> Therefore, for now, I maintain a rating of 4.
>
> [1] Carlini, N. and Wagner, D. Adversarial examples are not easily detected: Bypassing ten detection methods.

---

> > ### Author Response · Authors · 2024-08-12
> > **Thank you for the feedback.**
> >
> > Thank you for the feedback.
> >
> > (Minor 1) We believe that we have provided enough evidence that Robust networks (or Lipschitz networks) are not necessarily margin-consistent. In particular, Lipschitz does not imply margin consistency. However, we agree that the interplay between these concepts is an interesting direction for future research.
> >
> > (Major) We do not claim to solve adversarial examples detection. Instead, we aim at detecting (clean) samples on which the network’s decision is non-robust (susceptible of being attacked). When fixing the threshold on the logit margin to obtain a 95% True Positive Rate (for which the corresponding FPR are provided in Table 1), we observe that the logit margins on adversarial examples are almost always (much) smaller than that threshold. By doing this, we are not trying to differentiate between adversarial samples and clean samples which are close to the decision boundary. Instead, in this case, we both flag them as non-robust. As evidence of this, we provide the 99th percentile of the logit margin on adversarial examples together with the logit margin threshold at 95TPR in the table below.
> >
> > CIFAR10
> > | Model ID | Logit margin of Adversarial Examples (99 percentile) | Threshold of logit margin at 95% TPR for detection |
> > |---|---|---|
> > | AD1 | 0.01 | 1.42 |
> > | DS0 | 0.61 | 2.19 |
> > | DM0 | 1.64 | 3.01 |
> > | MR0 | 0.04 | 8.35 |
> > | MD0 | 0.06 | 11.04 |
> > | SE10 | 0.06 | 2.83 |
> > | PA20 | 0.14 | 1.69 |
> > | HE0 | 0.07 | 2.82 |
> > | TR0 | 0.07 | 4.11 |
> > | CL0 | 0.03 | 5.76 |
> > | EN0 | 0.13 | 5.06 |
> > | AL0 | 0.03 | 5.11 |
> > | AD20 | 0.03 | 1.33 |
> > | CU80 | 1.32 | 2.02 |
> > | RE802 | 0.13 | 1.87 |
> > | ZH0 | 0.17 | 2.87 |
> > | WA80 | 0.70 | 2.14 |
> > | AD10 | 0.01 | 1.20 |
> >
> > CIFAR100
> > | Model ID | Logit margin of Adversarial Examples (99 percentile) | Threshold of logit margin at 95% TPR for detection |
> > |---|---|---|
> > | AD1 | 0.01 | 1.42 |
> > | AD21 | 0.02 | 1.44 |
> > | CU81 | 0.13 | 2.09 |
> > | CU41 | 0.04 | 1.95 |
> > | DS1 | 0.13 | 1.36 |
> > | HE1 | 0.06 | 1.81 |
> > | PA21 | 0.08 | 1.56 |
> > | RA11 | 0.08 | 1.77 |
> > | RE812 | 0.17 | 1.73 |
> > | RE11 | 0.07 | 1.40 |
> > | RI1 | 0.03 | 3.42 |
> > | WA81 | 0.09 | 1.59 |
> > | WU1 | 0.06 | 2.02 |
> >
> > (Minor 2)  Figures 8 and 9 show, on a common y-axis, the distributions of distances between classifiers for each model. Comparing these ranges across models is not meaningful. We agree that these plots alone cannot provide evidence that equidistance holds in these models. In contrast, the results in Table 1 of the rebuttal PDF are evidence that the impact of the distance between classifiers is negligible for margin consistency; the logit margin and the feature margin lead to quite similar ranks. Thank you for pointing this out, we will modify section C of the supplementary material in this sense.

---

> ### Comment · Reviewer_WVo4 · 2024-08-12
>
> Thank you for the clarification, which cleared up my confusion on the two minor points. However, I would like to follow up on the possibility of adaptive attacks. I will raise my score if these questions can be sufficiently addressed.
>
> > We aim at detecting (clean) samples on which the network’s decision is non-robust (susceptible to being attacked).
>
> In this case, are the applications of the proposed method restricted to the case where 1) adversarial robustness is of interest; 2) clean examples are always provided?
>
> Could you please provide a motivating scenario that satisfies this assumption?
>
> > We observe that the logit margins on adversarial examples are almost always (much) smaller than that threshold.
>
> This is where an adaptive attack is important. With a white-box adaptive attack, is it possible to find adversarial examples (which trick the model into mispredictions) with larger logit margins? Is it possible to find examples that do not necessarily lead to mispredictions, but induce large logit margins without actually being far from the decision boundaries (i.e., lead to robustness overestimation)?
>
> Also, what types of attacks are used to generate the tables?
>
> > We are not trying to differentiate between adversarial samples and clean samples that are close to the decision boundary. Instead, we both flag them as non-robust.
>
> I agree that the proposed method can detect *clean* examples close to the decision boundary. However, for the above reasons, I am not fully convinced that adversarial examples can also be flagged.

---

> > ### Author Response · Authors · 2024-08-12
> > **Thank you once more for the reply**
> >
> > 1/ We believe it is interesting in the scenarios where, indeed, you would like to know the adversarial robustness of a sample (1) but may not afford to perform heavy adversarial attacks or use an even more expensive formal verification method. We agree that our results only provide empirical evidence that the logit margin is a good measure of the adversarial robustness of a clean IID sample (2).  Here are two applications that we have in mind:
> > * Given a large dataset, the logit margin can provide a reasonable estimate of the empirical robust accuracy by only performing attacks on a small subset of the dataset. The tables below show the robust accuracy estimates when attacking only 500 samples and compare them with the accuracies reported on the benchmark. The estimation is made by finding the logit margin threshold at 95%TPR for non-robust detection at $\epsilon=8/255$ on the subsample, then using that threshold to predict over the whole 10k test samples.
> > * In a real-time deployment scenario, the use of logit margin can be particularly beneficial if you know that your model is margin-consistent. In such a case, you can determine in real-time, just from the forward pass, which samples are vulnerable to adversarial attacks without actually performing an attack. This capability could be used for monitoring or making decisions, with the disclaimer that local robustness does not indicate whether the sample is wrong or not, and that the detection is not perfect. For instance, this could be important when the uncertainty of the camera sensor is known.
> >
> >
> > CIFAR10
> > | Model ID | Estimated Robust Accuracy | Reported Robust Accuracy |
> > |---|---|---|
> > | MD0 | 38.55 | 36.91 |
> > | MR0 | 39.61 | 39.12 |
> > | CL0 | 41.54 | 40.08 |
> > | AL0 | 40.94 | 40.21 |
> > | TR0 | 43.19 | 42.23 |
> > | EN0 | 50.91 | 49.25 |
> > | AD10 | 51.79 | 51.06 |
> > | AD20 | 53.42 | 52.48 |
> > | ZH0 | 53.87 | 53.08 |
> > | HE0 | 54.32 | 54.92 |
> > | SE10 | 56.00 | 55.54 |
> > | DS0 | 57.59 | 56.14 |
> > | DM0 | 59.77 | 57.27 |
> > | RE802 | 60.82 | 60.73 |
> > | PA20 | 63.04 | 61.04 |
> > | WA80 | 68.29 | 67.31 |
> > | CU80 | 69.01 | 67.73 |
> >
> > CIFAR100
> > | Model ID | Estimated Robust Accuracy | Reported Robust Accuracy |
> > |---|---|---|
> > | RI1 | 20.59 | 18.95 |
> > | AD1 | 27.51 | 27.14 |
> > | AD21 | 27.89 | 27.67 |
> > | HE1 | 29.56 | 28.42 |
> > | RE11 | 29.11 | 28.50 |
> > | WU1 | 29.38 | 28.86 |
> > | RA11 | 29.64 | 28.88 |
> > | PA21 | 31.69 | 31.08 |
> > | CU41 | 31.61 | 31.65 |
> > | RE812 | 33.11 | 32.06 |
> > | DS1 | 33.44 | 32.19 |
> > | WA81 | 38.96 | 38.83 |
> > | CU81 | 40.32 | 39.18 |
> >
> > 2/In the tables, we have used the adversarial attacks generated by the FAB attack. We acknowledge without reserve that adaptive attacks or other types of attacks may produce adversarial examples that can actually have bigger logit margins while being close to the decision boundary. We apologize if we cannot provide more results on this matter. While we have given it some thought, it is not clear to us at this point, how to generate such adversarial examples.

---

> ### Comment · Reviewer_WVo4 · 2024-08-13
>
> Thank you for the response.
>
> **Regarding the two applications:**
>
> > Given a large dataset, the logit margin can provide a reasonable estimate of the empirical robust accuracy by only performing attacks on a small subset of the dataset.
>
> ***This makes sense. Hence, I have increased my rating to 5.*** Please add this result to the paper.
>
> > In a real-time deployment scenario, you can determine in real-time, just from the forward pass, which samples are vulnerable to adversarial attacks without actually performing an attack.
>
> In a deployment scenario, attacks would be performed by some attacker outside the control of the deployer. Hence, the deployer may only receive adversarial examples and not have access to clean examples. Since the proposed method is only proven to work on clean examples, I am not convinced that it would be effective in this scenario.
>
> **Regarding FAB attack:**
>
> Is it the case that FAB attack finds minimally perturbed adversarial examples, and stops as soon as it finds one? If this is the case, then it makes sense why the output margins of adversarial examples are tiny. Does a similar phenomenon still hold for other  attack algorithms that do not terminate early, such as untargeted and targeted PGD?
>
> **Regarding adaptive attack:**
>
> > While we have given it some thought, it is not clear to us at this point, how to generate such adversarial examples.
>
> As mentioned above, PGD is an option. You can modify the attack objective, so that increasing output margin becomes part of the goal.
>
> You may also try AutoAttack. However, since the original AutoAttack also stops as soon as it finds an adversarial example, some modifications are needed to increase output margin (see *Minimum-Margin AutoAttack* in Appendix B.1 of [1] for an example).
>
> Regarding generating examples that lead to margin overestimation (but do not necessarily change the predicted class), one possibility is to run a PGD attack with an objective that maximizes output margin while restricting the distance from the nominal point (thereby restricting the input margin).
>
> [1] Bai et al. MixedNUTS: Training-Free Accuracy-Robustness Balance via Nonlinearly Mixed Classifiers.

---

> > ### Author Response · Authors · 2024-08-13
> >
> > Thank you for your response.
> >
> > * Regarding the deployment scenario, we agree that the model can receive all kinds of examples, including adversarial examples. For such instances, we confirm below on one of the strongly margin consistent models that adversarial examples crafted with standard attacks are indeed flagged as non-robust (in the same way as the non-robust clean examples) even if we cannot tell if they are adversarial or clean.
> >
> > * The standard AutoAttack sequentially runs the untargeted APGD-CE, the targeted APGD-DLR, the targeted FAB and the Square Attack. For evaluation, it stops as soon as it finds adversarial examples below the $\epsilon$ threshold. The latter attacks might not be run on all samples or not at all.  In order to get adversarial examples for each of these attacks, we ran the evaluations one by one and collected the adversarial examples produced.  We have also run the Carlini-Wagner attack (CW, cleverhans pytorch implementation) and the tentative adaptive attack using PGD that maximises the logit margin (PGD-MC). The table contains the 99 percentile value of the adversarial logit margins and shows that they are all less or equal to the logit margin threshold found to detect 95%TPR for non-robust detection (1.20, from the previously provided table).
> >
> > Model: AD10, $\epsilon=8/255$, 95TPR logit margin threshold=1.20
> > |  | APGD-CE | APGD-DLR | FAB | SQUARE | CW | PGD-MC |
> > |---|---|---|---|---|---|---|
> > | adversarial logit margin (99 percentile) | 0.928 | 0.779 | 0.004 | 0.204 | 0.004 | 1.230 |
> >
> > Even if the tentative adaptive PGD-MC finds bigger logit margins than other attacks, the 99th percentile is still quite at the limit (with 95th percentile = 1.07). Although this by no means dismisses the possibility of creating more effective adaptive attacks, it suggests that the task may not be as straightforward as initially perceived.
> >
> > * To find the minimally distorted adversarial for all samples, we used a FAB attack with a sufficient budget [1] and unbounded threshold, which does not stop the search as in the original AutoAttack.
> >
> > [1]Xu, Yuancheng, et al. "Exploring and exploiting decision boundary dynamics for adversarial robustness." arXiv preprint arXiv:2302.03015 (2023).

---

> > > ### Comment · Reviewer_WVo4 · 2024-08-13
> > >
> > > Thank you for the diligent response, and the result looks promising. It makes sense that FAB and CW, which aim to find minimally disturbed examples, result in small margins, whereas fixed-radius attacks such as APGD and Square find adversarial examples with larger margins, with the adaptive PGD-MC getting the largest output margin.
> > >
> > > Yes, AutoAttack stops early, but it is possible to modify it so that it doesn't. Nonetheless, I agree that this is too much to ask at this stage.
> > >
> > > If a comprehensive study like this were performed on multiple models in a rigorous format, I would have strongly advocated for acceptance. However, since the current results are only preliminary, **I maintain a rating of 5, which is still above the acceptance threshold**, because I agree that the other motivation (quick estimation of overall robustness) is valid.

---

### Official Review · Reviewer_utdZ · 2024-07-12

**Soundness:** 4
**Presentation:** 3
**Contribution:** 3
**Rating:** 7
**Confidence:** 4

**Summary:**

The authors propose using the distance between the two max values of a model's output as a proxy for the input margin to efficiently identify samples vulnerable to adversarial attacks. This proposed margin consistency is formally defined and shown to work across many robust models on the CIFAR10 and CIFAR100 datasets. Additionally, a basic learned fix is proposed for the few models which did not have margin consistency.

**Strengths:**

(S1) Proposed margin consistency for vulnerable sample detection in the context of robust models appears novel. Using logit statistics to detect adversarial examples is not new [1,3,4], and attacks have incorporated the difference between the two max logits [2]. However, this specific formulation (1), being on the defending side (2), and evaluating robust models (3) appear to be a novel combination.

(S2) Experiments are included for a wide variety of models, and a variety of metrics are provided for balanced evaluation. Additionally, the basic modification proposed for weakly correlated models significantly improves results.

(S3) Writing is clear, and mathematical formulation appears sound.

[1] Wang, Yaopeng, et al. "Model-agnostic adversarial example detection through logit distribution learning." 2021 IEEE International Conference on Image Processing (ICIP). IEEE, 2021.

[2] Weng, Juanjuan, et al. "Logit margin matters: Improving transferable targeted adversarial attack by logit calibration." IEEE Transactions on Information Forensics and Security 18 (2023): 3561-3574.

[3] Aigrain, Jonathan, and Marcin Detyniecki. "Detecting adversarial examples and other misclassifications in neural networks by introspection." arXiv preprint arXiv:1905.09186 (2019).

[4] Ozbulak, Utku, Arnout Van Messem, and Wesley De Neve. "Not all adversarial examples require a complex defense: Identifying over-optimized adversarial examples with IQR-based logit thresholding." 2019 International Joint Conference on Neural Networks (IJCNN). IEEE, 2019.

**Weaknesses:**

(W1) Additional experiments. Including experiments for one or two non adversarially robust models would help show the limitations (or additional potential) of margin consistency. Additionally, evaluation of a robust model trained on ImageNet would help show the scalability of the method. Pretrained robust ImageNet models are available [5], so if initial compute is not a factor computation should not be a hinderance.

(W2) No additional analysis for why margin consistency fails for the 2 CIFAR10 models.

[5] https://github.com/RobustBench/robustbench

**Questions:**

(Q1) What is the computational cost to build the dataset required for training in section 3.3? And in general, what is the computational cost required for the proposed method? Not as much of an issue due to adversarially robust models already requiring more compute to train and the proposed method taking minimal compute at inference.

(Q2) How successful would an adversarial attack maximizing the difference of the 2 max logits be in bypassing the proposed detection? Experiments for this may be slightly outside the scope of the paper.

**Limitations:**

yes

---

> ### Author Rebuttal · Authors · 2024-08-04
>
> Thank you for your review! Please see our comments in the general response about standardly trained models with zero adversarial accuracy and results on ImageNet. Below are our responses to your other concerns.
>
> **a/ (W2) No additional analysis for why margin consistency fails for the 2 CIFAR10 models.**
>
> We agree that further analysis of these two models may be helpful. However, we believe it may be closely related to a broader question on the relationship between robustness and margin consistency, which can be an avenue for future exploration. We have added an analysis in Figure 1 (PDF attached to the general response section) that gives further insights. It shows a disparity of correlations across different classes, while it is almost uniform in margin-consistent models (top row).
>
> **b/ (Q1) Computational cost.**
>
> Estimating input margins requires running the FAB attack with a sufficient budget [1]. The average time to estimate the input margins over 1000 samples is the following (on 16GB GPU-Titan XP):
>
> - CIFAR10: ResNet-18 (6min), ResNet-50 (19min), WideResNet-28-10 (35min)
> - CIFAR100: ResNet-18 (58min), ResNet-50 (7h), WideResNet-28-10 (6h)
>
> For ImageNet models, a ResNet-50 takes about 1 day 20hrs to 2 days on V100-32GB.
>
> **c/ (Q2) How successful would an adversarial attack maximizing the difference of the two max logits be in bypassing the proposed detection?**
>
> We believe it is possible to attack a sample to bypass that detection, but we think this is beyond the scope of this work. Nevertheless, in our case, we checked (line 178) that our non-robust samples are almost similar to the ones found by the standard AutoAttack evaluation which includes APGD-DLR that maximizes the DLR loss (Difference in Logits Ratio), which is a rescaled difference between the logit of the true class and the biggest logit among the others. This would seem to indicate that the logit margin can also detect vulnerable samples to this sort of attack.
>
> [1] Xu, Yuancheng, et al. "Exploring and exploiting decision boundary dynamics for adversarial robustness." arXiv preprint arXiv:2302.03015 (2023).

---

> > ### Comment · Reviewer_utdZ · 2024-08-12
> >
> > Thank you for the response. After reviewing the comments and concerns of other reviewers and your responses, I have decided to maintain my current rating.

---

### Official Review · Reviewer_jLJt · 2024-07-13

**Soundness:** 3
**Presentation:** 3
**Contribution:** 2
**Rating:** 6
**Confidence:** 3

**Summary:**

It is currently difficult to determine how susceptible a given input to a model is to adversarial perturbations. The distance from the input to the model's decision boundary in the input space (input space margin) is a reasonable metric, but it is intractible to compute for many deep neural networks and not always meaningful. This paper investigates the use of logit margin as a proxy for input margin. The logit margin is defined to be the difference between the two largest logits. Furthermore, a model is said to be margin consistent if there is a monotonic relationship between input margin and logit margins. Theorem 1 justifies that, if a model is margin consistent, then logit margin can be used to detect non-robust samples. Experiments show that common deep learning models can display a high level of margin consistency.

**Strengths:**

This work represents an interesting new perspective on detecting adversarial attacks. Rather than detecting adversarial examples themselves, the method presented here focuses on detecting when example is at risk of being attacked. This approach may lead be useful in the future in the design of new defense systems against adversarial attacks.

The experimental section of this work is strong. In particular, the correlation between input margin and logic margin of robust models is striking in figure 2 and serves as support for the plausibility of this metric.

The authors present a compelling hypothesis for when margin consistency might hold in machine learning models, relating margin consistency to isometry. This may lead to an interesting line of future work. A thoughtful commentary on directions for future work is also provided in Section 5.

**Weaknesses:**

While small logit margin may be associated with susceptibility to attacks, there may be other reasons we would not want to flag samples in this manner. For example, if a model exhibits differential performance on different subpopulations, certain subpopulations as a whole might be labeled as "brittle decisions." I think that an exploration and discussion of how pervasive brittle decisions are and how they are distributed in practice is warranted and would strengthen this paper.

Similarly, flagging brittle decisions may lead to a false sense of security. Adversarial examples have been shown to induce very high logit margins [1]. It is not clear to me that this method would be able to detect that an adversarially perturbed sample is close to a decision boundary. An enlightening experiment would be similar to figure 5a, except the samples have been adversarially perturbed to maximize logit margin.

[1] Goodfellow, Ian J., Jonathon Shlens, and Christian Szegedy. "Explaining and harnessing adversarial examples." arXiv preprint arXiv:1412.6572 (2014).

**Questions:**

- How would using a different distance measure change your results (i.e. measuring distance with $l_2$ norm rather than $l_\infty$ norm)?
- How do you think the experimental results would change if you were using a higher dimensional/more difficult to classify dataset? Would you expect the correlations observed in these results to hold?
- Do you think this method and your results have any implications for non-robust classifiers?

**Limitations:**

Limitations are discussed in section 5.

---

> ### Author Rebuttal · Authors · 2024-08-04
>
> Thank you for your review! Please see our comments in the general response for answers to your questions (Q1 and Q2 are addressed in the third point of the general response, and Q3 is addressed in the last point). Below are our responses to your other concerns.
>
> **a/ Bias across subpopulations**:
>
> Local robustness bias exists indeed in models; thank you for bringing up this interesting comment. Interestingly, when margin consistency (approximately) holds, we observe that robustness discrepancies across classes in input margin can also be observed using the logit margin (see top row, Figure 1 of the PDF attached to the general response section). We also observe that margin consistency (approximately) holds for each class individually, so no bias seems present as far as margin consistency goes. For the weakly margin-consistent models (example shown in the bottom row of Figure 1, attached PDF), there are significant disparities between the correlations across classes, which is also why using the logit margin for such models would be problematic.
>
> **b/ False sense of security and detection of adversarial examples:**
>
> Margin consistency, like standard accuracy, measures a property of the model over test inputs sampled iid from the training data distribution. From the objective they optimize, adversarial examples are close to the decision boundary and, therefore, have very small input margins, and we also observe that they have very small logit margins when compared to clean examples. However, we agree that it could, in principle, be possible for specially crafted adversarial examples to bypass margin consistency. This could be interesting for future work. The detection of adversarial examples itself is a different task, considered a defence strategy, and [1] shows that it is an equivalent task to robust classification. We believe studying margin consistency on adversarial examples or designing adaptive attacks is out of the scope of this work.
>
> [1] Tramer, Florian. "Detecting adversarial examples is (nearly) as hard as classifying them." International Conference on Machine Learning. PMLR, 2022.

---

> > ### Comment · Reviewer_jLJt · 2024-08-11
> >
> > Thank you for your response and for the new experimental results. My concerns have largely been addressed. However, I still believe that an investigation of adversarial/adaptive attacks would help the reader understand in what situations margin consistency is expected to hold. I have raised my score to a 6.

---

> > > ### Author Response · Authors · 2024-08-12
> > > **Thank you for your response**
> > >
> > > Thank you for your kind response and for raising your score. Note that we have provided some results on the logit margin on adversarial examples in our response to reviewer WVo4.

---

### Official Review · Reviewer_4V5x · 2024-07-30

**Soundness:** 3
**Presentation:** 3
**Contribution:** 4
**Rating:** 7
**Confidence:** 4

**Summary:**

The paper addresses the problem of efficiently detecting vulnerable inputs to a robust deep classifier at test time without the need for running adversarial attacks or formal verification. They introduce the idea of margin consistency of a classifier to connect the input-space margin and feature-space margin (or logit margin). A classifier is margin consistent if there is a monotonic increasing relationship between the input margin and logit margin. They show that margin consistency is a necessary and sufficient condition in order for the logit margin to be used as a score to detect (separate) "non-robust" (vulnerable) samples from "robust" samples.

Using a number of robust models trained using various adversarial training methods, from the RobustBench, they empirically show that a vast majortiy of these models exhibit high margin consistency (measured via a Kendall Tau correlation). For these models with high margin consistency, they also evaluate the detection power of the logit margin in separating non-robust samples from robust samples.

**Strengths:**

1. The paper address an important and practical problem of detecting vulnerable inputs to a robust deep classifier in an efficient way using the logit margin as a proxy for the input margin (when the model satisfies approximate margin consistency).

1. The development of ideas and presentation (with figures) is mostly clear and easy to follow.

1. The idea of connecting the input margin and logit (or feature space) margin via the margin consistency, and using it for detecting vulnerable (non-robust) inputs looks novel.

1. The experiments are extensive and explore interesting questions.

**Weaknesses:**

1. It's not fully clear why the paper does not use the feature-space margin directly instead of the logit margin. Taking the minimum of the distance to the decision boundaries (over classes $j \neq i$, where $i$ is the predicted class) in Equation 2 should give the feature-space margin. Why then is the logit margin, which is an approximation of this, needed? Because the equidistance assumption (line 123) may not hold in practice.

1. The proof of Theorem 1 is not precise and needs some clarifications (details in the "Questions" section).

**Questions:**

### Proof of theorem 1
**Sufficiency**: It seems like the finite sample $S$ assumption is not needed. \
Suppose $f_\theta(x)$ is margin consistent, and $A^S_\epsilon$ is the set of non-robust samples from $S$ for a given $\epsilon > 0$ (as defined in the paper).
Let
$\epsilon_0 = \sup\\{ d_{in}(\mathbf{x}) : \mathbf{x} \in A^S_\epsilon \\}$ and $\lambda_0 = \sup\\{ d_{out}(\mathbf{x}) : \mathbf{x} \in A^S_\epsilon \\}$.
For any non-robust sample $\mathbf{x} \in A^S_\epsilon$, its logit margin satisfies $d_{out}(\mathbf{x}) \leq \lambda_0$ due to the margin consistency. Therefore, $\lambda_0$ perfectly separates the non-robust samples from the robust samples based on the logit margin.
(This was a little unclear in the paper and does not need a finite $S$).

**Necessity**: It is important to state that the proof is by contradiction. Suppose that for any robustness threshold $\epsilon$, $d_{out}$ admits a threshold $\lambda$ that perfectly separates the non-robust samples from the robust samples. Assume that $f_\theta$ is *not* margin consistent. It seems to me that a three point example shows the contradiction better, since with two points, it is always possible to separate the points with the inequality direction reversed.

Since the model is not  margin consistent, there exist points $S = \\{ \mathbf{x}_1, \mathbf{x}_2, \mathbf{x}_3 \\}$ such that: i) in the input margin $d\_{in}(\mathbf{x}_1) < d\_{in}(\mathbf{x}_2) < d\_{in}(\mathbf{x}_3)$ and  ii) in the logit margin $d\_{out}(\mathbf{x}_1) < d\_{out}(\mathbf{x}_3) < d\_{out}(\mathbf{x}_2)$. Letting $\epsilon = d\_{in}(\mathbf{x}_2)$ (or slightly larger), we get the set of non-robust samples to be $A^S\_\epsilon = \\{ \mathbf{x}_1, \mathbf{x}_2 \\}$.

However, in this case, there exists no threshold $\lambda$ for the logit margin that can cleanly separate the non-robust samples  $\\{ \mathbf{x}_1, \mathbf{x}_2 \\}$ from the robust samples $\\{ \mathbf{x}_3 \\}$. Since this leads to a contradiction, the model must be margin consistent.


### Questions and Comments

1. In Eqn 2, $DB_j$ is not defined. I suppose it is $DB_j = \\{ \mathbf{z} \in R^m : (\mathbf{w}_j - \mathbf{w}_i)^T \mathbf{z} + b_j - b_i \geq 0 \\}$.

1. Referring to lines 123 -- 124, the equidistance assumption seems to be glossed over, and it is thereafter assumed that the robust deep classifiers satisfy the property? As mentioned under "Weaknesses", why can we not use the feature space margins?

1. Line 133: What does `output scores` refer to here? And clarify if it is the logit margin.

1. Line 136: It may be worth mentioning here that negative values of Kendall Tau (upto $-1$) imply that the two rankings are anti-correlated or reversed.

1. In section 2.3, the perfect discriminative function $g$ should be defined for a specific robustness threshold $\epsilon \geq 0$ and is a function of the classifier $f_\theta$. So it would be clearer to show this in the notation, e.g. as $g_\epsilon(\mathbf{x} ; f_\theta)$. Also the indicator function does not need the extra argument $\mathbf{x}$. That is, it can just be $\mathrm{1}\_{[d_{in}(\mathbf{x}) \leq \epsilon]}$

1. In Figure 3, it should be $x_2 \in A_{\epsilon_0}$, not  $x_2 \in A_{\epsilon}$. Also, on line 152, it should be $\lambda_0 = d_{out}(\mathbf{x}_0)$, to correspond to the figure.

1. The caption for Figure 4 is not so clear.

1. Lines 191, 192: the statement `... margin consistency is a property orthogonal to the robust accuracy ...` seems like a strong statement to make. It is possible that there could be a non-zero correlation (mutual information) between the two.

1. Figure 2 can be placed closer to the results section for readability.

1. Line 215: the word "boundary" is missing after "decision".

1. Based on line 220, it seems like a standard pointwise regression using MSE (and not learning to rank) is used to learn the mapping from the feature representation to a pseudo-margin. Could you also clarify why the Sigmoid activation is used at this network's output, for either regression or L2R?

1. Does the property of margin consistency and the detection of vulnerable samples extend to other norms such as $\ell_2$ norm?

1. Section 4, under OOD detection: the objective is usually to detect inputs that are from new classes not seen by the classifier during training, but it could also include detecting inputs that are from a different (shifted) marginal distribution, i.e., covariate-shifted OOD. It would be useful to cite a survey paper such as [Generalized out-of-distribution detection: A survey](https://arxiv.org/abs/2110.11334)

1. For mis-classification detection, a couple of relevant references are missing [2] and [3]. \
[2] https://papers.neurips.cc/paper_files/paper/2021/file/2cb6b10338a7fc4117a80da24b582060-Paper.pdf \
[3] https://openaccess.thecvf.com/content/CVPR2023/papers/Zhu_OpenMix_Exploring_Outlier_Samples_for_Misclassification_Detection_CVPR_2023_paper.pdf

**Limitations:**

Yes, there is discussion on the limitations and scope of this work in section 5.

---

> ### Author Rebuttal · Authors · 2024-08-04
>
> Thank you for your review! We will take the corrections into account. Please see our comments in the general response about defining the margin consistency in terms of the logit margin and results on the $\ell_2$ norm.
>
> **a/ The proof of Theorem 1.**
>
> Thank you for reading the proof carefully; we agree that there was some clumsiness in the sufficiency part. In this sense, we suggest modifying line 625 by:
>
> *"Let $x_0$ be an element of the finite set $A^S_\epsilon$ such that $d_{in}(x_0)=\max \{d_{in}(x): x\in A^S_\epsilon\}$ and let $\lambda = d_{out}(x_0)$."*
>
> We agree that formulating perfect separation for finite samples only is not fundamentally necessary. However, it appears to us that this opens the way for a simpler proof which avoids dealing with the intricacy of the continuum. We propose to add a comment in this sense in the paper.
>
> As for the necessity part, we need to indicate the proof is by contradiction, thank you for noticing this. We agree that our notion of perfect separability is oriented'' (i.e. non-robust points are required to be the ones with small $d_{out}$) and that we use this property in the proof. We understand that "it is always possible to separate the points with the inequality direction reversed", but this happens if and only if the model is "reversed margin consistent" ($d_{in}(x)<d_{in}(y)$ iff $d_{out}(x)>d_{out}(y)$), which cannot happen for the logit margin which is always non-negative and takes the value zero on that lie on input space decision boundaries (i.e. $d_{in}(x)=0$ implies $d_{out}=0$). While the three points counter example is attractive, it seems to us that the existence of such an example does not follow from the negation of margin consistency. We, therefore, suggest keeping this proof as it is while adding some comments that address the points you rightfully raised.
>
> **b/** $DB_j$  In Equation 2 should be the boundary $DB_{ij}$ defined similarly in the paragraph above
>
> $$\text{DB}_{ij} = \\{z'  \in  \mathbb{R}^m: (w_i-w_j)^\top z'+(b_i-b_j)=0\\}$$
>
> It is a good catch; we will correct it.
>
> **c/** Yes, in Line 133, we indeed refer to the logit margin as a score computed from the neural network's output (logits). We will clarify that.
>
> **d/ Lines 191, 192: the statement ... margin consistency is a property orthogonal to the robust accuracy ... seems like a strong statement to make.**
>
> We recognize that it is not the correct way to convey our message here, which is that being robust does not imply margin consistency (cf. discussion on Lipschitz smoothness vs. margin consistency, the first point of the answer to Reviewer WVo4). There may be indeed more to explore about the connection between the two properties. We propose to remove that statement and add the discussion about Lipschitz's smoothness and margin consistency.
>
> **e/ Pseudo-margin learning and the Sigmoid activation**
>
> The purpose of our study on pseudo-margin learning is to provide evidence that it is possible to learn to map the features  $h_\psi(x)$ to a proxy $d_{out}$ that simulates the margin consistency. Our approach is inspired by the setup used for confidence learning [1] (ConfidNet): a target confidence $y \in [0,1]$, Sigmoid as output activation with MSE loss. In our case, we use $y=\frac{d_{in}(x)}{\max \\{ d_{in}(z): z\in S\\}} \in [0,1]$ as a target, where S is the training set. In L2R, the earliest and simplest methods are point-wise methods, which are no different from standard regression with the MSE loss [2]. This worked worked well for our purpose.
>
> [1] Corbière, Charles, et al. "Addressing failure prediction by learning model confidence." Advances in Neural Information Processing Systems 32 (2019).
>
> [2] He, Chuan, et al. "A survey on learning to rank." International Conference on Machine Learning and Cybernetics. Vol. 3. Ieee, (2008).

---

> > ### Comment · Reviewer_4V5x · 2024-08-13
> > **Response to rebuttal**
> >
> > Thank you for your careful responses and additional results. I have read the other reviews and author responses. The points raised by Reviewer WVo4 regarding the method's applicability to adversarial inputs, adaptive attacks, and Lipschitz smoothness are important.
> >
> > I would encourage the authors to include some discussion on these aspects in the revised paper. Particularly, the method's potential vulnerability to carefully crafted adversarial inputs should be acknowledged as a limitation. Some of the results on adversarial inputs (from the review discussions) could also be included in the paper's appendices. The results on estimating the robust accuracy using the logit margin from a small test sample are interesting. These applications of the method could be included as motivation in the introduction (if not done already - I don't recall).
> >
> > Overall, I think the paper makes a valuable contribution towards a light-weight method for detecting non-robust or vulnerable samples at test time for robust models that satisfy margin consistency. Therefore, I will maintain my current score of 7 in favor of acceptance.

---

### Author Rebuttal · Authors · 2024-08-04

We sincerely thank the reviewers for their appreciation and thoughtful comments! The points raised by the reviewers are certainly very useful for clarifying and improving the presentation of our work while also bringing interesting avenues for future exploration. Below are some general comments about the margin consistency that are relevant to all reviewers. We individually respond to other reviewers' concerns right after.

1. Margin consistency is an order preservation property not implied by robustness. More precisely, Lipschitz's smoothness does not imply margin consistency—see details on that in the answer to reviewer WVo4. However, knowing how local robustness influences margin consistency and vice versa is an avenue for future exploration.

2. While equidistance may not be achieved perfectly in practice, we do not need perfect equidistance for the logit margin to be a practical approximation of the exact feature margin. Computing the exact feature margin requires computing the minimum over (K-1) pairs of scaled logit differences, where $K$ is the number of classes. The approximation circumvents the computational overhead of the minimum search, which can take a second instead of just microseconds for inference. This difference can add up to hours at scale, offering scalability when dealing with a large number of classes.  Table 1 in the attached PDF shows side-by-side the results when using the logit margin (Lm columns) and the exact feature margin (Fm columns). There is little to no difference between the two results.

3. Results for some ImageNet models in $\ell_\infty$ and $\ell_2$-robust models in Robustbench (Table 2, attached PDF file) show that these models are also strongly margin consistent. However, we do not exclude that exceptions may exist.

Note that measuring the $\ell_\infty$ norm for an $\ell_\infty$ robust model makes more sense than measuring another norm like $\ell_2$ because an $\ell_\infty$-robust model is not necessarily $\ell_2$-robust and vice-versa. However, the measured input margins in the $\ell_2$ norm and $\ell_{\infty}$ norm for the models we investigated are highly correlated, so the results do not change if we measure $\ell_2$ instead. We provide results in the cases where $\ell_2$ norm matters (i.e., for $\ell_2$-robust models, available only for CIFAR10 in Robustbench).

4. While we believe that margin consistency is far less attractive for standard models (for which almost all decisions are non-robust), we nonetheless provide Kendall $\tau$ correlations for a variety of standard models in the table below. Models can be margin-consistent without being robust and vice-versa. It is unclear at this point how we could leverage the margin consistency for non-robust models.

| Model ID              | Kendal tau | Accuracy | Architecture     | Dataset  |
|-----------------------|------------|----------|------------------|----------|
| STD\_M1               | 0.45       | 93.07    | ResNet18         | CIFAR10  |
| STD\_M2               | 0.69       | 92.60    | ResNet20         | CIFAR10  |
| STD\_M3               | 0.36       | 93.65    | ResNet-50        | CIFAR10  |
| STD\_M5 (Robustbench) | 0.50       | 94.78    | WideResNet-28-10 | CIFAR10  |
| STD\_M6               | 0.75       | 72.63    | ResNet-56        | CIFAR100 |
| STD\_M7 (Robustbench) | 0.70       | 76.52    | ResNet-50        | ImageNet |

---

### Decision · Program_Chairs · 2024-09-25

**Decision:**

Accept (poster)

**Comment:**

The reviewers agree that the paper tackles an important problem with sound results. The authors introduce the idea of margin consistency of a classifier to connect the input-space margin and feature-space margin (or logit margin). A classifier is margin consistent if there is a monotonic increasing relationship between the input margin and logit margin. They show that margin consistency is a necessary and sufficient condition in order for the logit margin to be used as a score to detect (vulnerable) samples from "robust" samples.